# Reinforcement Learning for Durable Algorithmic Recourse

## Abstract

Algorithmic recourse seeks to provide individuals with actionable recommendations that increase their chances of receiving favorable outcomes from automated decision systems (e.g., loan approvals). While prior research has emphasized robustness to model updates, considerably less attention has been given to the *temporal dynamics* of recourse—particularly in competitive, resource-constrained settings where recommendations shape future applicant pools. In this work, we present a novel time-aware framework for algorithmic recourse, explicitly modeling how candidate populations adapt in response to recommendations. Additionally, we introduce a novel reinforcement learning (RL)-based recourse algorithm that captures the evolving dynamics of the environment to generate recommendations that are both feasible and valid. We design our recommendations to be *durable*, supporting validity over a predefined time horizon $T$. This durability allows individuals to confidently reapply after taking time to implement the suggested changes. Through extensive experiments in complex simulation environments, we show that our approach substantially outperforms existing baselines, offering a superior balance between feasibility and long-term validity. Together, these results underscore the importance of incorporating temporal and behavioral dynamics into the design of practical recourse systems.

## 1 Introduction

Algorithmic recourse seeks to provide individuals who have been rejected by automated decision-making systems with counterfactual explanations that clarify the reasons for their rejection (Karimi et al., 2022; Rasouli & Yu, 2024; Rawal & Lakkaraju, 2020). These explanations typically consist of alternative feature values, close to the original ones, that would have led to a favorable decision (Wachter et al., 2017; Barocas et al., 2020).

Actionable recommendations based on counterfactual explanations enable individuals to improve their chances of acceptance in the future (Karimi et al., 2021; Upadhyay et al., 2025). However, *shifts in the training data, prediction model, or applicant pool can render such recommendations invalid over time, leading to situations where individuals who follow the suggested changes—often at significant time, labor, or financial costs—still get rejected* (Upadhyay et al., 2021; Fonseca et al., 2023). This issue of unreliable recourse is critical to address as it undermines trust in the system, may discourage individuals from engaging with it, and result in wasted effort (Rawal et al., 2021).

This concern has motivated the development of robust recourse methods that seek to remain effective in dynamic settings, contingent on the socio-technical context in which the system operates and responsive to the evolving conditions of the decision-making system and its environment (Upadhyay et al., 2021; Dominguez-Olmedo et al., 2022; Pawelczyk et al., 2023a). In particular, when considering *limited-resource, competitive* settings, it becomes essential to account for and manage the feedback effects of recourse on the applicant pool (Fonseca et al., 2023). Namely, as candidates repeatedly apply after attempting to follow the recommendations, the decision threshold may shift, potentially leading to a high rate of invalidity (Bell et al., 2024). While prior work has identified this issue and emphasized the limitations of existing recourse methods under such endogenous dynamics (Fonseca et al., 2023; Segal et al., 2024), *no comprehensive solution has yet been proposed*. We exemplify this gap in Appendix A with a motivating scenario on Ph.D. admissions.

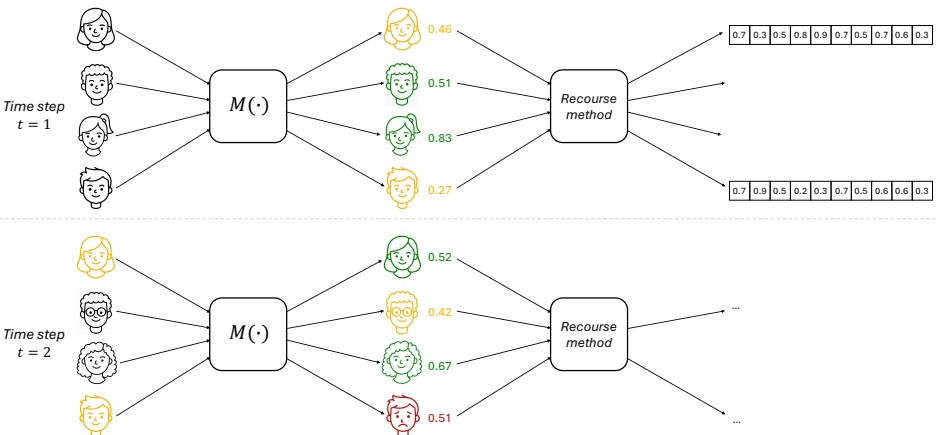

Figure 1: *Recourse invalidity*. At $t = 1$, four candidates apply, and the two with the highest scores are accepted. The decision threshold is $0.51$; following the state-of-the-art approach, rejected candidates (yellow) receive recommendations to reach this score. At $t = 2$, the rejected candidates from $t = 1$ (yellow) reapply, along with two new candidates (black). The yellow candidates have implemented the recommendations and raised their scores to around $0.51$. However, because of simultaneous recourse and a new candidate with a higher score, one reapplicant is still rejected.

In this work, we address this gap by modeling the problem through the lens of reinforcement learning (RL), interpreting the recommendation process as the policy of an RL agent, thereby capturing the sequential nature of interactions between the system and the applicants. The agent is trained to provide recommendations that are feasible, robust, and valid over a predefined time horizon $T$. Our contributions are as follows:

- We introduce a comprehensive, time-aware recourse framework that models a competitive, limited-resource setting in which recommendations are issued. Our environment captures varying feature-modification difficulties and delays between candidate reapplications, thus reflecting complex human behavior and contextual constraints.

- We propose a novel RL-based recourse algorithm that explicitly accounts for the feedback effects of recommendations on the applicant pool. To our knowledge, this is the first solution to the challenge of providing recourse in dynamic, resource-constrained environments. Our recommendations come with guarantees of validity over a configurable time horizon $T$, allowing candidates to delay reapplication while still benefiting from the same guidance.

- Through extensive experiments, we demonstrate the superiority of our method over the state-of-the-art, and we analyze how intrinsic context characteristics and stakeholder objectives shape the trade-off between the feasibility and validity of recommendations.

## 2 RELATED WORK

NEW

**Foundational Works on Algorithmic Recourse**. Algorithmic recourse emerged in response to concerns about the opacity of automated decision-making, particularly in the context of the GDPR's *right to an explanation*. A foundational contribution came from Wachter et al. (2017), who introduced counterfactual explanations as a way to help individuals understand and contest model outcomes; they casted recourse as an optimization problem, where the goal was identifying the smallest set of feature changes that would alter a decision. Building on this idea, Ustun et al. (2019) formalized recourse in terms of practical costs, proposing integer programming methods to generate actionable changes for linear classifiers. Subsequent work generalized these approaches to incorporate richer objectives (Dandl et al., 2020; Mothilal et al., 2020; Cheon et al., 2025; Rasouli & Yu, 2024; Rawal & Lakkaraju, 2020). Among these, Mothilal et al. (2020) introduced DiCE, which generates diverse sets of feasible counterfactuals.

A parallel line of research situates recourse within the framework of structural causal models (SCMs), emphasizing that feature dependencies constrain which interventions are feasible and

meaningful (Karimi et al., 2021; Beretta & Cinquini, 2023). Early work assumed access to the true underlying SCM (Karimi et al., 2021), whereas more recent methods seek to approximate the causal structure in practice (Karimi et al., 2020; Majumdar & Valera, 2024).                     NEW

**Dynamic recourse (exogenous shifts).** Another strand of work examines the robustness of recourse in dynamic environments (Altmeyer et al., 2023; Yang et al., 2025; Kayastha et al., 2024; Stundefinedpka et al., 2025; De Toni et al., 2025). Existing work has largely focused on *exogenous model shifts in non-competitive settings* (Upadhyay et al., 2021; Pawelczyk et al., 2023a; Guyomard et al., 2023; Nguyen et al., 2023). Upadhyay et al. (2021) proposed a min–max optimization framework that ensures recourse validity under worst-case perturbations to model parameters and inputs. Dominguez-Olmedo et al. (2022) introduced adversarially robust strategies for counterfactual generation, while Pawelczyk et al. (2023b) highlighted the trade-off between robustness and compliance with the right to be forgotten.                     NEW

**Reinforcement Learning solutions**. Recent research incorporates reinforcement learning into algorithmic recourse. For instance, De Toni et al. (2024) leverage RL to learn individual preferences and generate tailored recourse plans. Going further, Kanamori et al. (2025) apply RL to the concept of *improvement* (König et al., 2023), ensuring recommendations not only increase the chance of acceptance but also positively affect the system where the recourse is issued. Other work highlights the role of *risk* and imperfect user execution. To address this, Wu et al. (2024) use RL to balance cost and risk, providing policies that allow individuals to select safer options, while Xuan et al. (2025) use RL to generate robust action trajectories that account for imperfect execution. We note that none     NEW
of these works consider interactions between users or the effects of competition.                     NEW

**Competitive, limited-resourse setting**. Fonseca et al. (2023) and Bell et al. (2024) explore *endogenous population shifts* induced by recourse in *competitive environments*. They introduce an agent-based simulation framework to analyze how applicant competition affects recourse validity. They conclude that the state-of-the-art approach of pushing rejected candidates towards the last-seen decision threshold is ineffective, as it leads to high values of invalidity. While these works highlight the challenge of maintaining valid recourse under competition, they stop short of offering concrete solutions.

While prior work has significantly contributed to the field of algorithmic recourse, existing approaches primarily focus on improving individual recommendations. The literature, however, largely overlooks the endogenous feedback dynamics that arise in competitive environments with multiple candidates, where limited resources and strategic interactions continually reshape the decision boundary. This work addresses this critical gap, proposing a novel reinforcement learning method to generate feasible recourse recommendations that remain valid over a finite time horizon.

## 3 COMPETITIVE RECOURSE SETTING

In this section, we introduce the setting of the problem under study. We first describe the simulation environment in which candidates compete for a limited resource and modify their features based on recourse recommendations. We then formalize this environment as a reinforcement learning problem, where the objective is to identify an optimal policy for generating recommendations.

### 3.1 SIMULATION ENVIRONMENT

We build our time-aware recourse framework on prior work modeling recourse under limited resources and repeated applications (Fonseca et al., 2023; Bell et al., 2024), while introducing additional mechanisms to more thoroughly capture the dynamics of competitive recourse systems.

The simulation begins with an initial population $\mathcal{I}_0$ of $N_0$ candidates. More generally, we denote the population at time $t$ by $\mathcal{I}_t$, with size $N_t$. Each candidate $j$ is characterized by a feature vector $X_0^{\mathrm{F}}[j] \in [0,1]^z$, where $X_t^{\mathrm{F}} \in [0,1]^{N_t \times z}$ denotes the matrix of *factual* features for the candidate pool at time step $t$, and $z$ is the total number of features, that take values in $[0,1]$. Following prior work on competitive recourse (Fonseca et al., 2023; Bell et al., 2024), we assume that features are independently sampled from their respective marginal distributions, without causal dependencies among them. The details of the synthetic feature generation process are provided in Appendix B.

At each time step $t = 0, 1, 2, \ldots$, the population evolves as $m$ new candidates enter, $k$ candidates are accepted, and a variable number of candidates leave. A previously trained prediction model $M : [0,1]^z \to [0,1]$ assigns a qualification score to each feature vector. At each step, a threshold $th_t$ is chosen so that exactly $k$ candidates in $\mathcal{I}_t$ are accepted. For a candidate $j \in \{1, \ldots, N_t\}$ with features $X_t^{\mathrm{F}}[j]$, the acceptance indicator is defined as $h_k\big(M(X_t^{\mathrm{F}}[j]), th_t\big) \in \{0, 1\}$, where 1 denotes acceptance and 0 rejection.

Each rejected candidate $j$ is offered recourse in the form of a *counterfactual* feature vector $X^{\mathrm{CF}}[j]$, designed to ensure acceptance within $T$ time steps. Candidates decide whether to attempt the modification or exit the environment. This decision is governed by a *dropout probability*, which increases with both the number of failed attempts and the magnitude of required changes, modeling candidate discouragement (Grbic & Roskovensky, 2012).

For candidates who remain, each modification on each feature $i$ is implemented successfully with a *probability of success* that depends on the change magnitude, a feature-specific difficulty parameter $d_i \in [0, 1]$ (Lievens et al., 2005), and a global difficulty parameter $\beta$. In addition, each candidate has a *reapplication probability*, which increases with (i) *self-confidence*, measured by the extent to which recommended changes were applied, and (ii) *urgency*, determined by the time since the last application (Grbic & Roskovensky, 2012). Candidates may delay reapplication for up to $T$ steps, consistent with the guaranteed validity of the recommendation.

These extensions improve upon prior simulations (Fonseca et al., 2023; Bell et al., 2024), which assumed (i) zero dropout probability, meaning that candidates only left once accepted, (ii) uniform modification difficulty across features, and (iii) immediate reapplication without guarantees on recommendation duration. In other words, among the three key probabilities that we model, previous works only modeled the probability of success, making it dependent on the distance between the current score and the goal score, and a global difficulty parameter. By contrast, they treated the other two phenomena as deterministic: all rejected candidates reapply, and they do so at every available time step. We advance their framework by adding stochasticity to the modeling of reapplications and by modifying the probability of success, modeling one probability function per feature and making it depend on an additional parameter, the intrinsic feature difficulty. A detailed specification of these mechanisms is provided in Appendix B.

NEW

## 3.2 REINFORCEMENT LEARNING SETTING

We model the environment where the reinforcement learning agent is trained as a *Partially Observable Markov Decision Process (POMDP)*, capturing the sequential nature of algorithmic recourse under feedback loops, and extending the simulation framework introduced earlier.

Partial observability arises due to delays in candidate reapplications and exits. Individuals modify their features in response to prior recommendations, but these changes remain hidden until they reapply, if they do at all. Some may permanently exit the system due to discouragement, introducing further uncertainty into the environment.

Formally, the environment is a POMDP specified by the tuple $(\mathcal{S}, \mathcal{A}, \mathcal{P}, \Omega, O, R, \gamma)$, where $\mathcal{S}$ is the latent state space, $\mathcal{A}$ the set of actions, and $\mathcal{P}(s'|s, a)$ the transition function that defines the probability of moving from state $s$ to $s'$ after taking action $a$. The agent receives partial observations from an observation space $\Omega$, governed by the observation function $O(o|s', a)$, which defines the likelihood of observing $o$ upon reaching $s'$ via action $a$. The reward function $R(s, a)$ assigns a scalar signal to each state-action pair, and $\gamma \in [0, 1]$ is a discount factor balancing immediate and future rewards. We now describe the main components of the POMDP, starting from the latent state.

**State** $s_t$. The state $s_t$ captures the complete configuration of the environment at time $t$. It includes all candidates currently in the system, represented by their feature matrix $X_{c,t}$ and identifiers $\mathcal{I}_{c,t}$, as well as all candidates applying at this step—including new entrants—represented by $X_t^{\mathrm{F}}$ and $\mathcal{I}_t^{\mathrm{F}}$. Scores and binary outcomes for all candidates are obtained via the decision model $M(\cdot)$ and the acceptance indicator $h(\cdot)$. The state space $\mathcal{S}$ is continuous—since candidate features and scores are continuous—and its dimension varies with the number of candidates present and those reapplying.

**Action** $a_t$. The agent's action at time $t$ is defined as: $a_t = X_t^{\mathrm{CF}}$, where $X_t^{\mathrm{CF}}$ is a matrix of counterfactual feature vectors, each corresponding to a rejected candidate. These vectors represent the

feature configurations that, if adopted, would lead to acceptance, within a time window of $T$ steps. The action space $\mathcal{A}$ is continuous and of variable dimension.

**Transition Function** $\mathcal{P}(s_{t+1}|s_t, a_t)$. The environment evolves according to a stochastic transition function $\mathcal{P}$, mapping the current state $s_t$ and agent action $a_t$ to a distribution over successor states $s_{t+1}$. Transitions proceed in three phases. First, candidates with positive outcomes permanently exit the environment and are removed from $X_{c,t+1}$. Second, rejected candidates respond to their counterfactual recourse recommendations: some exit due to discouragement, while others remain and modify their features toward the suggested counterfactuals, updating $X_{c,t+1}$. Finally, a new application round occurs, comprising both new entrants and reapplying candidates previously rejected, forming the new feature matrix $X_{t+1}^{\mathrm{F}}$.

**Observation** $o_t$ **and Observation Function** $O(o_t \mid s_t, a_{t-1})$. The agent has partial observability of the environment, and the observation function specifies how this partial view is derived from the true latent state $s_t$ and the previous action $a_{t-1}$. Formally, the observation includes only the elements of $s_t$ corresponding to the current applicants: $o_t = (X_t^{\mathrm{F}}, \mathcal{I}_t^{\mathrm{F}})$. The observation space $\Omega$ is continuous and can have variable dimensionality depending on the number of applicants at time $t$.

**Reward Function** $R(s_t, a_t)$. The reward function integrates multiple objectives to ensure *equity*, *validity*, and *feasibility* of the agent's recommendations. To promote *equity*, we minimize disparities in the scores that rejected candidates would obtain if they implemented the recommended actions. Formally, we define the set of rejected candidates at time $t$ as $\mathcal{I}_t^{\mathrm{rej}}$. For each candidate $j \in \mathcal{I}_t^{\mathrm{rej}}$, the *goal score* is $g_t[j] = M(X_t^{\mathrm{CF}}[j])$, i.e., the score the candidate would achieve if they perfectly implemented the recommendation. We argue that these scores should be similar across rejected candidates, in order to prevent unequal treatments. Distinct from effort-based fairness (Bell et al., 2025), this reasoning aligns with merit-based fairness: candidates can receive recommendations that require different levels of effort according to their scores. This also discourages the agent from adopting degenerate strategies, such as implicitly encouraging some candidates to exit the environment by assigning them excessively difficult recommendations, thereby reducing competition for others. To guarantee this equity constraint, we minimize the *Gini index*:

$$\mathrm{Gini}_t = \frac{\sum_{i,j \in \mathcal{I}_t^{\mathrm{rej}}} |g_t[j] - g_t[i]|}{2 n_{r,t} \sum_{i \in \mathcal{I}_t^{\mathrm{rej}}} g_t[i]}, \tag{1}$$

where $n_{r,t} = |\mathcal{I}_t^{\mathrm{rej}}|$. Lower $\mathrm{Gini}_t$ indicates greater equity.

To ensure *validity*, we adopt the *Recourse Reliability* ($\mathrm{RR}_t$), first introduced by Fonseca et al. (2023), which measures the portion of candidates that successfully implemented a recommendation and were accepted at each time step:

$$\mathrm{RR}_t^T = \frac{|\mathcal{I}_t^{\mathrm{succ}} \cap \mathcal{I}_t^{\mathrm{acc}}|}{|\mathcal{I}_t^{\mathrm{succ}}|}. \tag{2}$$

where $\mathcal{I}_t^{\mathrm{succ}}$ indicates the candidates that successfully implemented a recommendation and reapplied at time step $t$, and $\mathcal{I}_t^{\mathrm{acc}}$ indicates the candidates accepted at step $t$. In the original formulation, $\mathcal{I}_t^{\mathrm{succ}}$ included only candidates reapplying from the previous step. We extend this to candidates whose last application was within the past $T$ steps, so $\mathrm{RR}_t^T$ measures reliability over a $T$-step horizon.

To prevent trivial solutions that maximize $\mathrm{RR}_t^T$ by suggesting extremely difficult modifications, we introduce the *Recourse Feasibility* ($\mathrm{RF}_t^T$), which quantifies the fraction of candidates who received recommendations within the past $T$ steps and reapplied with a successful implementation at time $t$:

$$\mathrm{RF}_t^T = \frac{|\mathcal{I}_t^{\mathrm{succ}}|}{|\mathcal{I}_{t-T:t}^{\mathrm{rej}}|}, \tag{3}$$

where $\mathcal{I}_{t-T:t}^{\mathrm{rej}}$ is the set of candidates who last applied unsuccessfully in the window $[t - T, t - 1]$, and thus could have reapplied at $t$, with a perfectly implemented recommendation. In this way, the metric penalizes failed implementations, delays, and discouragement-related exits.

**Policy** $\pi(a_t|s_t)$. The agent learns a policy $\pi(a_t|s_t)$ that defines a distribution over recommendations $a_t$, conditioned on the current environment state $s_t$. Learning this policy is challenging due to the high-dimensional, variable-sized state and action spaces. In the next section, we introduce a training framework that mitigates the computational burden associated with these large and dynamic spaces.

## 4 Reinforcement Learning Solution

NEW

In the previous section, we interpreted the simulation environment through the lens of reinforcement learning, framing the recourse task as learning a policy that yields valid, durable, and feasible recommendations. In this section, we describe the strategy used to search for the optimal policy within this environment.

Directly learning the full counterfactual matrix $X_t^{\text{CF}}$ is computationally expensive due to its high dimensionality and variable size. To address this, we adopt a hierarchical approach that separates counterfactual generation from goal selection, explicitly modeling the dependency between the two.

**Counterfactual generation**. We first learn a stochastic function

$$\phi : (x_t^{\text{F}}, g) \mapsto \text{Dist}(x_t^{\text{CF}}),$$

that defines a probability distribution over counterfactual feature vectors $x_t^{\text{CF}}$, conditioned on a candidate's features $x_t^{\text{F}}$ and a target score $g$. Samples from this distribution are required to satisfy $M(x_t^{\text{CF}}) \approx g$ while minimizing a cost function that measures the discrepancy between $x_t^{\text{F}}$ and $x_t^{\text{CF}}$. In other words, $\phi$ specifies how to probabilistically modify features to achieve a desired score.

**Goal selection policy**. Given the pre-trained $\phi$, we learn a stochastic policy

$$\mu : (X_t^{\text{F}}, \mathcal{I}_t^{\text{F}}) \mapsto \text{Dist}(g_t),$$

that defines a probability distribution over target scores $g_t$. During training, the pre-trained $\phi$ translates sampled goal scores into actionable recommendations for each rejected candidate:

$$g_t \sim \mu(X_t^{\text{F}}, \mathcal{I}_t^{\text{F}}), \quad X_t^{\text{CF}}[j] \sim \phi\big(X_t^{\text{F}}[j], g_t\big), \quad \forall j \in \mathcal{I}_t^{\text{rej}}.$$

The environment evolves according to these recommendations, making the training of $\mu$ inherently dependent on $\phi$.

In this hierarchical setup, $\mu$ decides *what score to aim for*, while $\phi$ determines *how to modify the features* to reach that score. Pre-training $\phi$ reduces the computational complexity and stabilizes the training of $\mu$. While $\mu$ is primarily responsible for the trade-off between *Recourse Reliability* (Equation 2) and *Recourse Feasibility* (Equation 3) in the reward function, the Pareto efficiency of this trade-off largely depends on $\phi$, as the feasibility of a recommendation critically depends on the trajectory taken to reach the target score. Additionally, $\phi$ indirectly optimizes the Gini index (Equation 1), by providing recommendations that approximately lead to the same score for all candidates.

This two-step architecture mirrors state-of-the-art recourse methods, which typically fix the goal score $g_t$ at the last-seen decision threshold and optimize only $\phi$. Our approach instead learns $g_t$ adaptively, based on the behavior of the candidates. We further design $\phi$ to improve the balance between reliability and feasibility, targeting higher values of both metrics. Concretely, $\phi$ is implemented as an RL policy, the *recourse recommender policy*, pre-trained with respect to the target-score policy, the *predictor policy*. We next describe the training procedure for both agents.

### 4.1 Training of the Recourse Recommender Policy

The recourse recommender policy $\phi$ is trained in a simplified environment derived from the setting introduced earlier. The state at time $t$ is defined as $s_t = (x_t^{\text{F}}, g)$, where $x_t^{\text{F}}$ is the feature vector of a single candidate and $g$ a target score. The action is the counterfactual feature vector $a_t = x_t^{\text{CF}}$. Although the deployment setting of the policy remains unchanged, with respect to the setting presented in Section 3.2, the POMDP used for training is modified to pursue a different objective, which is achieving a predefined target score, and to focus exclusively on a single candidate, which is sufficient for the intended task. This reduction significantly lowers the computational burden.

NEW

Training proceeds over multiple episodes. At the start of an episode, a goal score $g$ is sampled such that $M(x_0^{\text{F}}) < g$. At each step, the agent proposes a recommendation $x_t^{\text{CF}}$, which the candidate attempts to implement. Each recommendation has validity $T = 1$, meaning candidates reapply at every step. The episode terminates when $M(x_t^{\text{F}}) \geq g$ or a maximum number of steps is reached.

Recommendations are evaluated on two criteria: (i) *accuracy:* how closely $M(x_t^{\text{CF}})$ approaches $g$, and (ii) *cost:* the effort required to modify $x_t^{\text{F}}$ into $x_t^{\text{CF}}$. The accuracy objective ensures that

the recourse recommender can generate paths toward arbitrary targets, and that—once paired with the recommender—it leads to counterfactual scores that are consistent with goal scores, thereby improving the Gini-based reward (Equation 1). Formally, accuracy is measured as

$$e_t = |M(x_t^{\text{CF}}) - g|. \tag{4}$$

The cost objective encourages minimal-effort modifications. We define an estimated cost function that penalizes large changes and prioritizes easier-to-modify features, based on estimated difficulties:

$$\hat{c}_t = \sum_{i=1}^{z} |x_t^{\text{CF},(i)} - x_t^{\text{F},(i)}| \cdot \hat{d}_i, \tag{5}$$

where $z$ is the number of features, and $\hat{d}_i$ is the agent's estimate of the difficulty of modifying feature $i$. Difficulty estimates are learned adaptively; the full procedure is detailed in Appendix C.

For optimization, we employ the Soft Actor-Critic (SAC) algorithm (Haarnoja et al., 2018), a model-free, off-policy method well-suited for continuous action spaces. In our work, $\phi$ is trained *online*, interacting directly with the environment; the same procedure can also be executed *offline* if a sufficiently rich dataset, related to another recourse system employed in this setting and containing information on how candidates respond to recommendations, is available.  FIX

### 4.2 Training of the Predictor Policy

The predictor policy, denoted by $\mu$, is trained on the POMDP introduced in the previous section. Within the hierarchical framework, the action space is reduced to $a_t = g_t$, i.e., the selection of a target score. During training, the recourse recommender policy $\phi$ is treated as a fixed component of the environment: it provides the counterfactual updates required to construct $X_t^{\text{CF}}$, based on $g_t$, while $\mu$ focuses solely on learning how to select appropriate goals. The reward function used to train $\mu$ excludes the Gini term, as it is entirely handled by the recourse recommender policy.

Because the environment is only partially observable and the reward is non-Markovian, we augment both the state and observation spaces with explicit historical information. At each time step $t$, the agent receives a window of data covering all candidates who applied and were rejected during $[t - T, t - 1]$. For each such candidate, the following metadata are provided: (i) their feature vector at the time of their last application, (ii) their unique identifier, (iii) the time step of their most recent application, (iv) the total number of applications they have submitted, (v) the most recent recourse recommendation received. This represents information that the recourse system, represented by the  NEW agent, has seen at some point; we assume that such data can be stored. By explicitly including these variables in the agent's observation, rather than requiring it to infer or internally store past events, we ensure that the environment is fully Markovian with respect to the predictor's decision process. This design choice facilitates stable learning in the presence of delayed effects.

Training is conducted over fixed-length episodes. At the beginning of each episode, a new population of initial applicants is generated. The predictor $\mu$ is then optimized using SAC (Haarnoja et al., 2018), chosen for its sample efficiency and ability to handle continuous action spaces.

## 5 Experiments

### 5.1 Setup

In this section, we present the performance evaluation of our method relative to established baselines from the literature. Additionally, we analyze how environmental constraints and design choices influence achievable performance. Our approach is compared against three widely used baselines for non-causal recourse (Ustun et al., 2019; Wachter et al., 2017; Mothilal et al., 2020) (hereafter called Ustun, Wachter, and DiCE). Some of these methods have also been adopted as baselines in recent studies on recourse under competition (Fonseca et al., 2023; Bell et al., 2024); in our framework, they serve as alternatives to the recourse recommender policy $\phi$.

Each strategy is combined with: (i) a *trivial predictor*, which applies the classifier's most recent decision threshold (reflecting standard practice in dynamic recourse), and (ii) our proposed predictor, parameterized by policy $\mu$. This yields two categories of methods: (i) *baselines*, pairing each recourse strategy with the trivial predictor, and (ii) *hybrids*, pairing the strategies with our predictor.  NEW

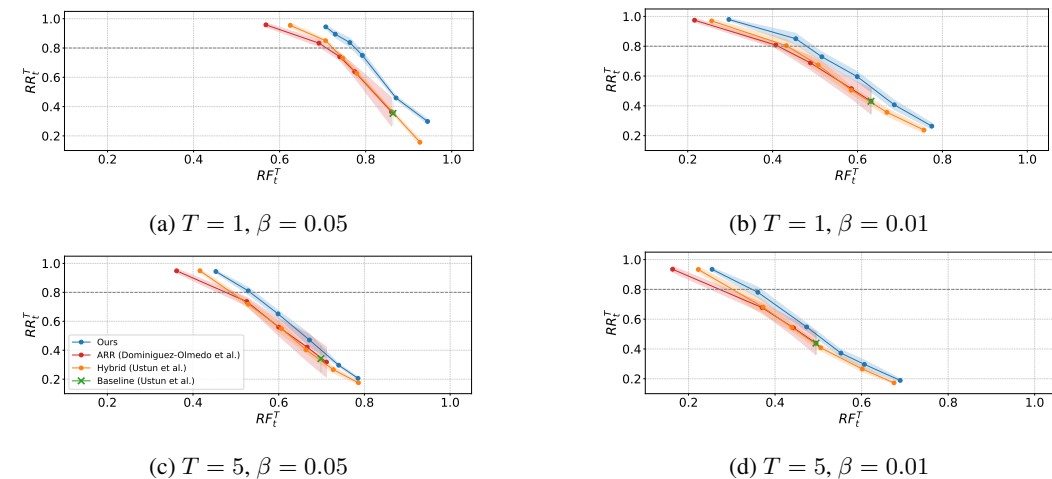

(a) $T = 1$, $\beta = 0.05$        (b) $T = 1$, $\beta = 0.01$

(c) $T = 5$, $\beta = 0.05$        (d) $T = 5$, $\beta = 0.01$

Figure 2: Comparison of Pareto fronts of our method (blue), the hybrid method based on Ustun's approach (orange), the baseline using Ustun's approach (green), and the ARR method (red), across four settings with $T \in \{1, 5\}$ and $\beta \in \{0.05, 0.01\}$. Pareto fronts plot Recourse Reliability $\mathrm{RR}_t^T$ and Recourse Feasibility $\mathrm{RF}_t^T$, each averaged over ten evaluation episodes. The gray line at $RR_t = 0.8$ denotes the *high reliability threshold*, distinguishing configurations that achieve desirable recourse reliability.

We further compare our approach to the method of Dominguez-Olmedo et al. (2022), referred to as Adversarially Robust Recourse (ARR). ARR accounts for adversarial perturbations to an individual's features; Since $M(\cdot)$ is linear and the features are independent, the ARR objective reduces to adjusting the target score according to $\varepsilon$. Counterfactuals for reaching this adjusted target score are then derived using the procedure of Ustun et al. (2019). We evaluate ARR across a range of $\varepsilon$ values.

We evaluate all methods under four experimental conditions, varying the recourse horizon ($T \in \{1, 5\}$) and the setting difficulty ($\beta \in \{0.05, 0.01\}$). The reward coefficients ($\alpha, \tau$), and the robustness coefficient $\varepsilon$, which govern the trade-off between Recourse Reliability and Recourse Feasibility, are chosen to produce Pareto frontiers spanning recourse reliability values $\mathrm{RR}_t^T$ approximately in $(0.20, 0.95)$.

Details on training and evaluation are illustrated in Appendix D. Since baselines tend to overlap, we depict only Ustun and ARR for plot readability. Analogous results, including comparisons with Wachter and DiCE, and analysis of the Gini Index of all methods, are provided in Appendix E.

## 5.2 RESULTS

Figure 2 shows Pareto plots for Recourse Feasibility ($\mathrm{RF}_t^T$) and Recourse Reliability ($\mathrm{RR}_t^T$) across all four experimental settings, averaged over ten evaluation episodes. Regarding our method and the hybrid approach, each point on a Pareto front corresponds to a different trained predictor $\mu$ with varying values of the parameters $\alpha$ and $\tau$. In the case of ARR, each point corresponds to a different value of $\varepsilon$. The horizontal gray line at $RR_t = 0.8$ indicates the *high reliability threshold*, highlighting simulations that achieve a desirable level of recourse reliability.

**Impact of the time horizon** $T$. Comparing the top and bottom plots in Figure 2, we observe that the value of $T$ strongly affects the validity-feasibility trade-off: achieving high validity requires policies with lower feasibility as $T$ increases. Guaranteeing recourse over a longer horizon imposes a more stringent requirement, forcing the agent to recommend more challenging feature changes.

Figure 3 further highlights this phenomenon, by plotting the average Recourse Feasibility $\mathrm{RF}_t^T$, fixing $\mathrm{RR}_t^T = 0.95$ and $\beta = 0.05$, for $T \in [1, 5]$. As noticed, feasibility must decrease to guarantee large reliability over an increasing time horizon $T$.

One additional challenge of a longer time horizon is slower convergence, highlighted in Figure 4. It presents the convergence curves of two predictor agents trained under identical conditions ($\beta =$

0.01, $\alpha = 7$, $\tau = 5$), but with different planning horizons: $T = 1$ and $T = 5$. Each point represents the cumulative reward averaged over the previous ten episodes.

For $T = 1$, the reward begins to increase after a few hundred episodes and converges to a final value of $\approx 600$ after about 2000 steps. In contrast, for $T = 5$, the reward starts improving only after roughly 1000 episodes and reaches a final value of $\approx 400$ after around 3000 steps. This behavior shows that the agent requires substantially more exploration when validity must be guaranteed over a longer horizon, since the task is more complex.

**Impact of the setting difficulty** $\beta$. Comparing the left and right panels of Figure 2 shows that the value of $\beta$ strongly shapes the attainable trade-off between $\mathrm{RF}_t^T$ and $\mathrm{RR}_t^T$.

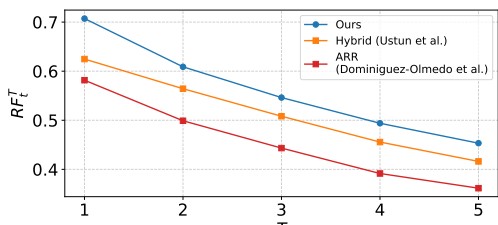

Figure 3: Recourse Feasibility $\mathrm{RF}_t^T$, for a fixed value of Recourse Reliability $\mathrm{RR}_t^T$ ($\approx 0.95$), and $\beta = 0.05$, varying $T \in [1, 5]$, for our method, the hybrid (based on Ustun's approach), and ARR.

We recall that $\beta$ scales the probability of successfully implementing feature modifications: higher values correspond to higher probabilities of success, while lower values make modifications more difficult. In both scenarios, to ensure high reliability, the agent recommends relatively high target scores, that push reapplying candidates above new applicants. For large $\beta$, this strategy has a moderate negative impact on $\mathrm{RF}_t^T$, since even challenging modifications remain feasible. In contrast, for low $\beta$, the same strategy yields a much sharper trade-off, as many candidates are unable to realize the recommended changes.

This analysis reveals another intrinsic limitation of recourse in resource-constrained environments. When the means for improvement are inherently difficult (low $\beta$), it is challenging to devise recommendations that are both likely to be implemented and sufficient to guarantee a positive outcome. Consequently, practitioners must carefully prioritize among these desiderata.

Recourse based on Wachter and DiCE highlights the same trends, as shown in Appendix E.

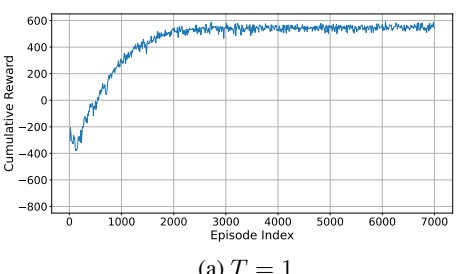

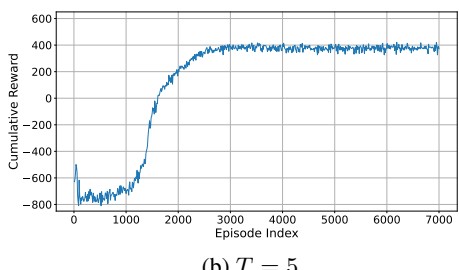

(a) $T = 1$   (b) $T = 5$

Figure 4: Convergence curves in two identical settings ($\beta = 0.01$, $\alpha = 7$, $\tau = 5$), comparing $T \in \{1, 5\}$. The y-axis shows the average cumulative reward (smoothed over ten episodes), and the x-axis denotes the episode index.

**Comparison of baseline, hybrid, ARR, and our approach**. Across all subplots in Figure 2, the baseline approach (green) attains low reliability ($\mathrm{RR}_t^T \approx 0.4$) while favoring feasibility. This outcome reflects the limitations of simply using the last observed decision threshold as the target score instead of tailoring it to an evolving environment. In contrast, our predictor $\mu$ can be plugged into any recourse recommender, such as Ustun, explicitly governing this trade-off (orange). These results highlight the advantage of an RL-based predictor over a simplistic fixed threshold policy. Comparing ARR (red) to the baseline (green), we observe that robustifying the target score based on a fixed parameter $\varepsilon$ improves recourse reliability, at the cost of reduced feasibility. However, the hybrid method (orange) Pareto-dominates ARR in all four settings, particularly in high-reliability regimes ($RR_t$ above the high reliability threshold). This demonstrates that environment-aware target adjust- NEW

ments outperform naive approaches that ignore the candidates' behavior. A more in-depth analysis of this is provided in Appendix F.

Figure 2 also compares our method (blue) with the hybrid approach (orange). Our approach achieves Pareto optimality across all four experimental settings. The key distinction lies in the recommendation strategy: while Ustun's method selects recommendations based solely on minimal feature changes, our policy $\phi$ explicitly accounts for feature modification difficulties, prioritizing changes to easier features, thus resulting in more feasible recourse paths. Hybrid methods based on Wachter and DiCE achieve a performance similar to Ustun, as illustrated in Appendix E.

We further note that the relative advantage of our method depends on the setting parameters. When the complexity of the problem—encoded in the parameters $\beta$ and $T$—increases, the Pareto front of our method moves closer to that of the hybrid. One possible interpretation is that, while in favorable conditions ($T = 1$, $\beta = 0.05$) more attainable recourse paths directly correspond to a higher portion of implementing candidates (for the same validity), in more constrained environments (e.g., $T = 5$ or $\beta = 0.01$) this mapping is less immediate, limiting the achievable gain.

Even under these stricter conditions, however, our method continues to provide robust improvements and maintains good trade-offs. This indicates that our approach remains effective over longer horizons $T$, highlighting its practical advantage in dynamic, multi-step settings.

## 6 DISCUSSION AND CONCLUSIONS

This paper presents the first solution to the problem of *robust recourse recommendations in competitive, limited-resource settings*. Our approach leverages reinforcement learning to anticipate candidate responses to recommendations and to generate suggestions that jointly maximize feasibility and validity. By adaptively estimating the relative difficulty of modifying each feature, the method prioritizes more accessible changes. Moreover, it supports recourse validity for $T$ time steps, where $T$ is specified by the stakeholder issuing the recommendations.

While the RL agent effectively learns environment dynamics, real-world deployment may introduce additional complexities. A key drawback arises during the transient learning phase—especially for the recourse recommender policy—where candidates may receive suboptimal recommendations. This limitation could be mitigated by training this policy on offline data, such as data from previous recourse systems used in the same setting, which record how the candidate population evolved in response to recommendations. Alternatively, the system could be pre-trained in our simulation environment, using domain expertise to align it with the actual setting; once online training on real data begins, such pre-training would help avoid fully sacrificing performance on transient candidates while allowing faster adaptation and convergence. Exploring such extensions constitutes an important step toward practical deployment and is left for future work.  NEW

Moreover, our simulation environment focuses on non-causal recourse, reflecting the work's emphasis on robustness under competition. The predictor, however, can be paired with any causal recourse method, and the recommender can be trained in environments with underlying causal structures. Additional sources of uncertainty, such as shifts in new applicants' distribution or in the prediction model, could also be modeled (Appendix B.5). Exploring these extensions provides a promising path for validating robustness and optimality in more complex, causally grounded settings.  NEW

Lastly, while our framework empirically shows that RL can be leveraged in competitive recourse settings to find a solution that balances multiple desiderata, no theoretical guarantees of convergence to such a solution can be established, as the chosen RL method (Haarnoja et al., 2018) lacks such guarantees (Appendix H).

Overall, this work establishes a foundation for durable and adaptive recourse under competition, while opening multiple pathways for further research.

## ETHICS STATEMENT

Our work addresses algorithmic recourse, a line of research that seeks to empower individuals affected by automated decisions by identifying feasible changes they can undertake to alter future

outcomes. While this vision has clear ethical appeal, it is important to acknowledge several limitations and potential risks.

First, recourse recommendations can inadvertently *shift responsibility* from institutions to individuals, obscuring systemic sources of unfairness (Sullivan & Kasirzadeh, 2024). A model may suggest behavioral changes (e.g., increasing income or reducing debt), but such recommendations risk deflecting attention from structural inequities encoded in data and decision pipelines.

Moreover, many recommended changes may not map cleanly to real-world actions or may implicitly require resources unequally available across social groups. This raises concerns about feasibility, fairness, and inclusivity (Barocas et al., 2020).

Lastly, explanations and recourse operate in *adversarial contexts*, where institutions and affected individuals may have misaligned incentives (Bordt et al., 2022). In such settings, post-hoc recourse mechanisms are vulnerable to manipulation, selective disclosure, and explanation hacking, which can undermine their transparency and accountability.

By highlighting these limitations, we aim to situate our contribution responsibly. Our research does not directly involve human subjects or sensitive personal data, but it engages with concepts that can influence downstream applications in high-stakes domains. We encourage future work to complement technical advances in recourse with attention to their social, legal, and institutional implications.

## REPRODUCIBILITY STATEMENT

We ensure reproducibility by providing a repository with our code in the supplementary material. Mathematical details of the environment are presented in Appendix B. A comprehensive description of our approach is given in Section 4 and Appendix C. Finally, Appendix D reports the complete specifications of the experimental setup.

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

## A  MOTIVATING EXAMPLE

To demonstrate the limitations of existing recourse methods, we examine a Ph.D. admission process. Decisions on admissions are supported by a screening system $M(\cdot)$ that evaluates applicants using criteria such as their GPA, educational background, publications, awards, extracurricular activities, English proficiency, and admission test scores. Admission to the next stage is granted to the top $k$ applicants, where $k$ remains a constant value representing the number of seats available annually.

The goal is to provide rejected candidates with actionable recommendations—feature changes likely to lead to future acceptance, such as, for example, "Upgrade your education from Bachelor's to Master's", or "Increase your test score from 65% to 70%". The motivation for this goal is highlighted by Venkatasubramanian & Alfano (2020): recourse is a fundamental right, and people should be empowered to reverse impactful algorithmic decisions through feasible actions.

State-of-the-art methods typically generate recourse by identifying feature changes that bring a rejected applicant's score to the current threshold. However, this approach can fail in competitive settings. For example, in Figure 1, at time $t = 1$, two candidates are accepted and two rejected. Recommendations are given to the rejected candidates to reach the threshold score of 0.51. But at $t = 2$, after implementing these changes, a candidate is still rejected, since more than $k = 2$ candidates now meet or exceed the previous threshold. This occurred because the recommendation did not account for the increased competition caused by the recourse itself.

This results in wasted effort, financial cost, and loss of trust in the system. The candidate has acted on the recommendation expecting acceptance, only to be denied again. The issue lies in generating overly easy recommendations that too many can follow, leading to more applicants being able to implement them than available slots. To address this, we propose an approach that anticipates population-level responses and selects more robust target scores. The goal is to ensure that only a subset of candidates can reach these targets, guaranteeing acceptance for those who do. At the same time, recommendations must remain *feasible and actionable*.

Moreover, we introduce the concept of *feature-modification difficulty*, a measure of how difficult it is to change a feature, to reflect real situations constraints. For example, to reach a predefined target score, a candidate might either:

1. Publish a first-author paper at a top-tier conference, or

2. Improve English proficiency from B2 to C1 and increase their test score from 65% to 85%.

While the second option requires more changes, it may be more preferred, as the first option requires resources that the student may not have, and entails a high level of uncertainty. Since precise difficulty ratios are rarely known in advance, we propose estimating them by observing candidate behavior over time.                                                                                      NEW

Finally, recommendations must consider long implementation times and reapplication delays. Following Venkatasubramanian & Alfano (2020), we argue that recourse should either be permanent or come with an explicit expiration date. Offering candidates recommendations that are only valid for a single time step risks creating a false sense of agency, since they may have no realistic way to implement the required changes within that interval. We adopt the latter option and associate each recommendation with a *validity horizon $T$*, during which the recommendation ensures acceptance. This allows candidates to plan longer-term changes with confidence that their efforts will remain relevant.

In real deployments, practitioners could select $T$ by combining domain knowledge with empirical evidence. Domain expertise can help estimate the typical time required for individuals to implement meaningful changes to key features, allowing stakeholders to choose a horizon that balances recommendation validity with practical feasibility. Additionally, if historical data were available on applicants who received recommendations and later reapplied, one could empirically estimate the distribution of reapplication intervals or the time needed to implement specific changes.

We note that although the responsibility to ensure longer durability may not be as strong as the responsibility to ensure validity (since the latter corresponds more directly to breaking a promise), both are tied to user trust and to the system's broader accountability. Failures on either front can lead

candidates to disregard the recommendations altogether, ultimately rendering the recourse system ineffective.

## B    SIMULATION ENVIRONMENT DETAILS

### B.1    SYNTHETIC DATASET GENERATION

To train the predictive model $M(\cdot)$ for estimating candidates' qualification levels, we construct a synthetic dataset of 10,000 examples, each characterized by 10 continuous features. These examples represent past candidates who were either accepted or rejected. Each feature is independently sampled from a normal distribution, with its mean and standard deviation drawn from uniform distributions, to introduce variability across features. All features are subsequently normalized to lie in $[0, 1]$.

Labels are designed to reflect subjective and occasionally inconsistent human decision-making. Specifically, a weighted sum of the features is computed using randomly assigned weights sampled from $[0.1, 1]$ and normalized to sum to 1. Gaussian noise with mean 0 and standard deviation 0.05 is added to this score. Candidates with a score exceeding 0.5 are labeled as accepted; all others are labeled as rejected.

The same feature generation procedure is applied to produce candidate populations $\mathcal{I}_0$ at the beginning of each episode, and new applicants at each time step. In this case, ground-truth labels are not generated, as they are unnecessary for the simulation.                                                                    NEW

Our design extends previous simulation environments modeling competitive recourse in limited-resource settings (Fonseca et al., 2023; Bell et al., 2024). Prior work considered candidates generated in a 2-dimensional feature space, sampled independently at random, where $x = (x_1, x_2)$ and $x_i \sim \mathcal{N}(\mu = 0.5, \sigma = 0.3)$ for $i = 1, 2$. They trained a simple logistic regressor as a classifier, with target variables $y_i$ drawn from a binomial distribution.                                                                    NEW

We improve this synthetic data generation process by considering 10 continuous features instead of 2, with each feature sampled from a distinct Gaussian. Moreover, we train the logistic regressor on a dataset constructed in this way, where the ground-truth target is correlated with the features, while still incorporating noise.

### B.2    DROPOUT PROBABILITY

The likelihood of a candidate dropping out depends on two factors: the gap between their current score and the goal score, and the number of previous applications. Intuitively, candidates are more likely to withdraw when they are far from the goal or have already reapplied multiple times.

Formally, let $b_j = \max(0, g - M(X^{\mathrm{F}}[j]))$ denote the distance of candidate $j$'s score from the goal score $g$, and let $q_j$ be the number of reapplications submitted up to time step $t$. The dropout probability is modeled as a function of these variables, with three decay coefficients: $\rho$ (effect of the score gap), $\chi$ (effect of reapplications), and $\omega$ (their interaction).

$$p_{\mathrm{dropout}} = 1 - \exp(-(\rho b_j + \chi q_j + \omega b_j q_j)). \qquad (6)$$

This exponential form ensures that $p_{\mathrm{dropout}}$ increases monotonically with both $b_j$ and $q_j$, approaching 1 as either grows large. Conversely, when $b_j = 0$ and $q_j = 0$, the dropout probability is minimized at $p_{\mathrm{dropout}} = 0$, corresponding to a candidate already meeting the goal score on their first attempt.

The term inside the exponent, $\rho b_j + \chi q_j + \omega b_j q_j$, can be interpreted as a *discouragement factor*, jointly capturing how performance shortfall and repeated failures contribute to disengagement.

### B.3    PROBABILITY OF SUCCESSFUL IMPLEMENTATION

For a candidate $j$ with features $X^{\mathrm{F}}[j]$, the probability of successfully implementing a recommended change on feature $i$ depends on:

- the amplitude of the recommended change, $|X^{\mathrm{CF},i}[j] - X^{\mathrm{F},i}[j]|$,

- the feature modification difficulty $d_i \in [0, 1]$,
- the target value $X^{\text{CF},i}[j]$, and
- the global scaling parameter $\beta$, which controls the overall difficulty of feature changes.

We note that the explicit dependence on the target value reflects the intuition that reaching extreme goals is more challenging, even when the starting point is close.

We define the *attainability* of feature $i$ for candidate $j$ as:

$$a_{j,i} = \frac{1}{|X^{\text{CF},i}[j] - X^{\text{F},i}[j]| \cdot X^{\text{CF},i}[j]} - 1. \tag{7}$$

Attainability is minimized at 0 when $|X^{\text{CF},i}[j] - X^{\text{F},i}[j]| = X^{\text{CF},i}[j] = 1$, and diverges to infinity when any of the denominator terms approaches zero. Intuitively, $a_{j,i}$ quantifies the feasibility of implementing a specific feature change.

The probability of success is then modeled as:

$$p_{\text{success}} = 1 - \exp\left(-\beta \cdot \frac{a_{j,i}}{d_i}\right), \tag{8}$$

where higher $\beta$ increases the likelihood of success across all features. This probability lies in $[0, 1]$ and increases monotonically with attainability. Specifically, when $|X^{\text{CF},i} - X^{\text{F},i}| = d_i = X^{\text{CF},i} = 1$, we obtain $p_{\text{success}} = 0$, while if any of these terms is zero, the probability approaches 1.

### B.4   PROBABILITY OF REAPPLYING

At each time step, a candidate's decision to reapply depends on two factors: *self-confidence*—the extent to which they have implemented the recommendation—and *urgency*—the time elapsed since their last application.

We model the reapplication probability as a convex combination of a distance-based base probability and a time-based scaling factor.

The base probability measures the candidate's alignment with the goal score. For candidate $j$, it is defined as:

$$p_{\text{base},j} = \exp(-\nu \cdot b_j), \tag{9}$$

where $\nu$ is a decay parameter, and $b_j$ is the distance of the candidate's current score to the goal score, as previously defined.

The time-based factor captures the increasing tendency to reapply as time passes:

$$u_j = \frac{t - l_j}{T}, \tag{10}$$

where $t$ is the current time step, $l_j$ the last application time step, and $T$ the recourse validity horizon.

The final probability of reapplication is:

$$p_{\text{reapply},j} = (1 - u_j) \cdot p_{\text{base},j} + u_j. \tag{11}$$

This formulation guarantees that $p_{\text{reapply},j}$ increases monotonically with time and converges to 1 either when $u_j = 1$ (i.e., after $T$ steps since the last application) or when $p_{\text{base},j} = 1$ (i.e., the recommendation has been perfectly implemented).

### B.5   POSSIBLE ENVIRONMENT EXTENSIONS                                        NEW

Our simulation environment is intentionally designed to be extensible, allowing richer behavioral dynamics to be incorporated as needed. In this way, future work could capture more complex candidate behaviors and assess the agent's ability to learn an effective policy in the presence of additional sources of noise.

Table 1 summarizes several phenomena that could be included, together with indicative implementation strategies.

| Phenomenon | Implementation strategy |
|---|---|
| Heterogeneous urgency and self-confidence | Introduce personalized parameters in $p_{\text{reapply}}$, sampled per candidate. |
| Collective action among candidates | Assign a small probability (e.g., 0.03) to a coordinated *non-engagement* event. If the event occurs, set $p_{\text{success}} = 0$ for every rejected candidate and set $p_{\text{reapply}} = 1$ at the next time step. |
| A priori low trust in the recourse system | For each candidate receiving a recommendation, sample a low-probability event (e.g., 0.05). If the event occurs, set that candidate's $p_{\text{success}} = 0$, regardless of the attainability (for all the time-steps where such candidate reapplies). |
| Exogenous variation in candidate features | Sample a low-probability event (e.g., 0.05). If the event occurs, select a subset of features and shift the mean or standard deviation of their sampling distributions. |
| Exogenous shifts in the predictive model | With small probability (e.g., 0.05), retrain the model on a modified dataset where the weights for computing the ground truth are slightly perturbed. |

Table 1: Potential extensions to the simulation environment and corresponding implementation strategies.

Each phenomenon requires modifications to specific components of the environment. For instance, personalization of candidates' urgency or self-confidence can be reflected by introducing per-candidate parameters in the reapplication model, scaling the exponent in Equation 9 and $u_j$ in Equation 10.

Collective action can be modeled as a rare global event in which candidates strategically choose not to pursue recommendations to avoid intensifying competition. Under such an event, candidates follow their usual dropout dynamics via $p_{\text{dropout}}$, but all remaining candidates reapply at the first available time step with unchanged features (i.e., $p_{\text{success}} = 0$ and $p_{\text{reapply}} = 1$).

Even when collective action does not occur, candidates may individually decline to engage due to low trust in the recourse system. These candidates similarly keep their features unchanged and reapply at the next opportunity (i.e., $p_{\text{success}} = 0$ and $p_{\text{reapply}} = 1$), regardless of the attainability of their recommendations.

Exogenous shifts may arise in both the candidate population and the predictive model $M(\cdot)$. Shifts in candidate features may be implemented by perturbing the sampling distributions of selected features, modifying their means or standard deviations. For model shifts, retraining $M(\cdot)$ on a perturbed version of the synthetic training set (whose ground-truth weights are slightly altered) provides a simple mechanism to modify the relationship between features and outcomes, thereby affecting the model's learned weights.

## C  REINFORCEMENT LEARNING SOLUTION DETAILS

### C.1  FEATURE DIFFICULTIES ESTIMATION

To estimate feature difficulties, we assume partial knowledge of the environment—specifically, the parametric form that links feature difficulties to the probability of successfully implementing a recourse action. Without loss of generality, we fix the parameter $\beta$ as known. As indicated in Equation 8, $\beta$ acts only as a scaling factor on the difficulties, controlling the overall level of difficulty in the simulation. Consequently, if $\beta$ were unknown, it could be absorbed into the difficulty parameters $d_i$ and estimated jointly with them.

Initially, all estimates of feature difficulties are set to $\hat{d}_i^{(0)} = 0.5$, for all features $i$. After each recourse attempt, we observe whether each feature change was successfully applied, for the only candidate in the environment. Let $y_i^{(t)} \in \{0, 1\}$ denote this binary outcome (at time $t$), and let $p_i^{(t)}$ be the predicted probability of success, based on the current belief on $\hat{d}_i$. We then compute the error signal:

$$err_i^{(t)} = (p_i^{(t)} - y_i^{(t)}) \cdot a_i^{(t)}, \tag{12}$$

which represents the discrepancy between predicted and observed outcomes, scaled by the attainability $a_i^{(t)}$ (previously introduced).

Feature difficulties are updated using a decaying learning rate:

$$\hat{d}_i^{(t+1)} = \text{clip}\Big(\hat{d}_i^{(t)} + \eta_i^{(t)} \cdot err_i^{(t)}, \, 0, \, 1\Big), \tag{13}$$

where

$$\eta_i^{(t)} = \frac{\eta_0}{1 + V_i^{(t)}}, \tag{14}$$

with $\eta_i^{(0)} = 0.05$ as the base learning rate and $V_i^{(t)}$ the number of prior updates to feature $i$. The clipping ensures that updated difficulties remain within $[0, 1]$.

This online procedure allows the model to iteratively refine its estimates of feature difficulties based on observed behavioral responses to counterfactual recommendations.

### C.2 RECOURSE RECOMMENDER TRAINING

The recourse recommender is trained in a simplified environment with a single candidate. The reward penalizes both the error in Equation 4 and the cost in Equation 5.

To facilitate learning, the reward evolves in two phases. During an initial warm-up period, it depends only on the error term, enabling the agent to learn accurate mappings toward predefined goals. Once feature-modification difficulty estimates stabilize, the cost term is introduced. From this point, the agent operates in a constrained RL setting, where it must choose the *lowest-cost* recommendation among those that reach the target.

The combined reward is:

$$r_t = \begin{cases} -\varphi \cdot c_t, & \text{if } e_t \leq \varepsilon, \\ -\varphi \cdot c_t - \psi \cdot (e_t - \varepsilon), & \text{otherwise,} \end{cases} \tag{15}$$

where $\varepsilon$ is a tolerance threshold, and $\varphi, \psi$ are hyperparameters with $\psi \gg \varphi$.

Over training, both the difficulty estimates $\mathbf{d}$ and the recommendation policy converge, yielding a recourse recommender capable of producing accurate and low-cost counterfactuals.

## D EXPERIMENTAL SETUP

The first step in our experimental setup is to construct the score-based decision model $M(\cdot)$. We generate a synthetic dataset of $10\,000$ candidates, each described by 10 features and a binary ground-truth label indicating past acceptance or rejection. $M(\cdot)$ is a logistic regression model, trained on this dataset to approximate the ground-truth labels. The model's probabilistic outputs serve as candidate scores, representing the estimated likelihood of acceptance.

The same data generation procedure is used to initialize candidate instances for training the policy $\phi$. Training episodes for the recourse recommender span up to 10 time steps and are conducted in two phases. In the first phase, the reward is based solely on prediction error (Equation 4), and training runs for 3,000 episodes. In the second phase, the reward incorporates both prediction error and modification cost (Equation 15), and training continues for an additional 20,000 episodes. The parameters used are $\varepsilon = 0.01$, $\varphi = 10$, and $\psi = 300$.

The predictor policy is trained on a simulated population initialized with $N = 20$ candidates. At each time step, $k = 9$ candidates are accepted and $m = 10$ new candidates are introduced. The

feature difficulties are set to $\mathbf{d} = [0.84, 0.15, 0.85, 0.78, 0.25, 0.18, 0.29, 0.83, 0.91, 0.10]$. Each episode comprises 100 time steps. The reward function for the predictor is defined as:

$$R(s_t, a_t) = \alpha \cdot \left(1 + 0.90 \cdot \log(\mathrm{RR}_t^T)\right) + \tau \cdot \left(1 + 0.90 \cdot \log(\mathrm{RF}_t^T)\right), \tag{16}$$

where the logarithmic transformation emphasizes the impact of low values of both metrics. The coefficients $\alpha$ and $\tau$ are positive and adjusted across simulations. The predictor is trained for 7,000 time steps.

# E ADDITIONAL RESULTS

## E.1 RECOURSE RECOMMENDER POLICY PERFORMANCE

The summed absolute error between the true difficulties $\mathbf{d}$ and their estimates $\hat{\mathbf{d}}$ is given by

$$e_{\mathrm{diff}} = \sum_{i=1}^{z} |d_i - \hat{d}_i|, \tag{17}$$

and is approximately $3 \times 10^{-2}$, indicating high fidelity in the difficulty estimation process.

After training the recourse recommender, we assess its performance using the prediction error from Equation 4 and the *true* modification cost:

$$c_t = \sum_{i=1}^{z} |x_t^{\mathrm{CF},(i)} - x_t^{\mathrm{F},(i)}| \cdot d_i, \tag{18}$$

where, relative to Equation 5, the estimated difficulties $\hat{d}_i$ are replaced with their true values $d_i$. Both quantities are averaged over ten evaluation episodes.

Our method achieves an average error of $1.9 \times 10^{-3}$ and an average cost of $5.9 \times 10^{-2}$ (Table 2).

For comparison, we applied the same protocol to Ustun, Wachter, and DiCE. Ustun's method achieved near-zero error ($e_t = 2.2 \times 10^{-16}$) but incurred a substantially higher cost ($c_t = 3.0 \times 10^{-1}$). Wachter and DiCE obtained errors of the same order as our method but at high costs, similarly to Ustun.

The strong precision of Ustun's method is expected: it employs integer programming to compute exact minimal changes for achieving the target score in linear models. In contrast, Wachter's, DiCE's, and our RL-based approach rely on approximate, gradient-based or learning-based optimization. Consequently, they exhibit slightly higher error values but remain applicable to a broader class of models, unlike Ustun's approach which is restricted to linear formulations.

| Method | $e_t$ | $c_t$ |
|--------|-------|-------|
| Ours | $1.9 \times 10^{-3}$ | $5.9 \times 10^{-2}$ |
| Ustun | $2.2 \times 10^{-16}$ | $3.0 \times 10^{-1}$ |
| Wachter | $2.6 \times 10^{-3}$ | $2.7 \times 10^{-1}$ |
| DiCE | $1.6 \times 10^{-2}$ | $3.6 \times 10^{-1}$ |

Table 2: Average prediction error and modification cost—computed with respect to the *true* feature difficulties—for our recourse recommender $\phi$ and the comparison approaches. Each method is evaluated under conditions matching the training setting of our recourse recommender, and results are averaged over ten evaluation runs.

These results highlight the effectiveness of the proposed policy in balancing fidelity to the target decision with minimizing modification cost.

## E.2 COMPARISON WITH WACHTER AND DICE

Figure 5a compares our method with Wachter in a setting with $\beta = 0.05$ and $T = 1$, while Figure 5b analogously compares DiCE. As observed previously, the baseline achieves low values of reliability ($\approx 0.4$ for Wachter and $\approx 0.6$ for DiCE), prioritizing feasibility. On the other hand, the hybrid variant provides greater control over the trade-off between feasibility ($\mathrm{RF}_t^T$) and validity ($\mathrm{RR}_t^T$), achieving high validity ($\mathrm{RR}_t^T \approx 0.95$) while maintaining feasible recommendations ($\mathrm{RF}_t^T \approx 0.60$), in both cases. Overall, the Pareto fronts of the hybrid variants closely match that shown in Figure 2a, related to Ustun's approach. Our method remains Pareto-optimal, identifying more attainable paths to reach a target score.

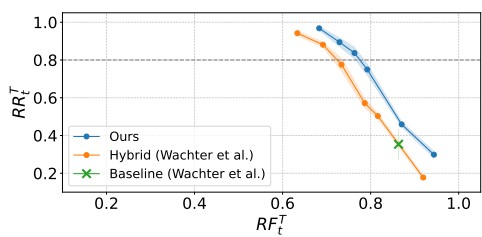
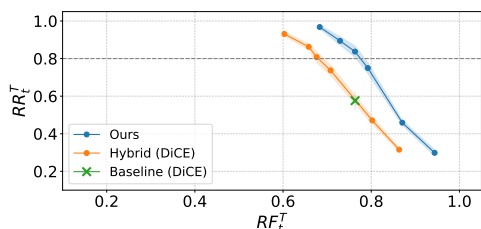

(a) Comparison with Wachter's approach.

(b) Comparison with DiCE.

Figure 5: Comparison of Pareto fronts of our method (blue line), the hybrid method based on Wachter's approach and DiCE (orange line), and the baseline method using Wachter's approach and DiCE (green dot), in a setting with $T = 1$ and $\beta = 0.05$. The Pareto fronts plot the Recourse Reliability $\text{RR}_t^T$ (averaged over ten evaluation episodes) against the Recourse Feasibility $\text{RF}_t^T$ (also averaged over ten evaluation episodes).

| | Gini Index | |
|---|---|---|
| **Method** | $T = 1$ | $T = 5$ |
| Ours | $3.2 \times 10^{-3}$ | $3.0 \times 10^{-3}$ |
| Ustun | $2.3 \times 10^{-16}$ | $2.3 \times 10^{-16}$ |
| Wachter | $1.8 \times 10^{-4}$ | $1.8 \times 10^{-4}$ |
| DiCE | $1.3 \times 10^{-2}$ | $1.2 \times 10^{-2}$ |

| | Recourse Feasibility $\text{RF}_t^T$ | |
|---|---|---|
| **Method** | $\beta = 0.05$ | $\beta = 0.01$ |
| Ours | $0.71 \pm 0.01$ | $0.30 \pm 0.02$ |
| Ustun | $0.63 \pm 0.03$ | $0.26 \pm 0.02$ |
| Wachter | $0.61 \pm 0.02$ | $0.30 \pm 0.02$ |
| DiCE | $0.55 \pm 0.02$ | $0.25 \pm 0.03$ |

Table 3: Comparison of Gini indices. Results are averaged over ten episodes and reported for two settings ($T = 1$, $T = 5$). All methods are matched on average $\text{RR}_t^T \approx 0.95$ and $\beta = 0.05$.

Table 4: Recourse feasibility ($\text{RF}_t^T$), averaged over ten evaluation episodes, for a fixed recourse reliability ($\text{RR}_t^T \approx 0.95$) and $T = 1$, across different values of $\beta$.

### E.3 Analysis of the Gini Index of Each Recourse Recommender

We evaluate the average Gini index and recommendation cost of our policy $\phi$ as well as the methods by Ustun, Wachter, and DiCE, when paired with our predictor $\mu$. The evaluation considers both $T = 1$ and $T = 5$. For each recourse recommender, we train a dedicated predictor, ensuring comparability by selecting models that achieve an average Recourse Reliability, over ten evaluation episodes, of approximately $0.95$. The Gini index (Table 3), defined in Equation 1, is computed over ten evaluation episodes.

The results indicate that varying $T$ has no substantial effect on the Gini index. As expected, Ustun's method yields extremely low values ($\approx 10^{-16}$), reflecting near-perfect equity. The other methods achieve higher but still reasonably low values. This behavior aligns with our earlier discussion (Section 4): a recourse recommender that more precisely maps to a predefined score produces lower dispersion in target scores, and hence a lower Gini index. Accordingly, the observed indices are strongly correlated with the average errors reported in Table 2. Importantly, while Ustun's method achieves the greatest precision, our approach delivers equitable recommendations that, as demonstrated in the main text, also attain high feasibility and reliability.

### E.4 Impact of $\beta$ on the Validity-Feasibility Trade-off

We analyze the effect of $\beta$ on the balance between validity and feasibility. Lower values of $\beta$ correspond to settings in which feature changes are more difficult to implement, making the trade-off between maintaining high validity ($\text{RR}_t^T$) and achieving feasible recourse ($\text{RF}_t^T$) more pronounced.

Table 4 shows that decreasing $\beta$ substantially worsens the validity-feasibility trade-off. While validity is held fixed ($\text{RR}_t^T \approx 0.95$), feasibility drops sharply: for example, our method's $\text{RF}_t^T$ falls from 0.707 at $\beta = 0.05$ to 0.365 at $\beta = 0.01$. This highlights that even strong methods face limited options in stringent settings, making the balance between feasible and validity particularly challenging in such settings.

## F    IN-DEPTH COMPARISON OF OUR APPROACH, THE HYBRID (BASED ON USTUN ET AL.) AND ARR (DOMINIGUEZ-OLMEDO ET AL.)

NEW

| | $T = 1$ | | $T = 5$ | |
| --- | --- | --- | --- | --- |
| **Method** | $\beta = 0.05$ | $\beta = 0.01$ | $\beta = 0.05$ | $\beta = 0.01$ |
| **Ours** | $0.71 \pm 0.01$ | $0.30 \pm 0.02$ | $0.45 \pm 0.01$ | $0.25 \pm 0.01$ |
| **Hybrid (Ustun et al.)** | $0.63 \pm 0.03$ | $0.26 \pm 0.02$ | $0.42 \pm 0.03$ | $0.22 \pm 0.02$ |
| **ARR (Dominiguez-Olmedo et al.)** | $0.57 \pm 0.02$ | $0.22 \pm 0.02$ | $0.36 \pm 0.02$ | $0.16 \pm 0.01$ |

Table 5: Recourse feasibility ($\text{RF}_t^T$), averaged over ten evaluation episodes, for a fixed recourse validity ($\text{RR}_t^T \approx 0.95$), varying $T \in \{1, 5\}$ and $\beta \in \{0.05, 0.01\}$.

In Table 5, we zoom into the results of our approach, the hybrid variant based on Ustun et al. (2019), and the ARR approach (Dominguez-Olmedo et al., 2022) in high-reliability regimes (i.e., $RR_t = 0.95$). As shown in the table, our method achieves higher feasibility at the same reliability level than the hybrid approach, by prioritizing changes on features with lower difficulties. Meanwhile, the hybrid approach, which uses our recommender to choose a target score based on the environment's characteristics and candidates' behavior, outperforms the ARR method, whose recommendations consist of a target score derived from a robustified threshold (parameterized by $\varepsilon$) at each time step.

The higher feasibility observed in this case is likely due to the fact that, while ARR robustifies the threshold at every time step, our recommender predicts future competition. As a consequence, in some cases it anticipates a decrease in competition in subsequent time steps and therefore lowers the target score.

Specifically, when examining the behavior of our trained recommender $\mu$ (paired with Ustun) in a short evaluation episode (Table 6) with $\beta = 0.05$ and $T = 1$, we observe that there are time steps in which the agent recommends a target score lower than the most recent threshold ($t = 2$ and $t = 4$). These recommendations still yield high levels of reliability. As a result, the average recourse feasibility $RF_t$ increases, since candidates are only recommended strong changes when necessary.

Instead, when examining the behavior of ARR in a short evaluation episode (Table 7), we observe that, as expected, the score recommended by ARR is always higher than the most recently observed threshold. As a consequence, the values of recourse feasibility $RF_t$ are generally lower.

## G    EXPERIMENT ON GERMAN

NEW

To assess the suitability of our method in real-world settings, we conducted an additional experiment using the German Credit dataset[1]. Although information on how candidates respond to recommendations is not available in this dataset—a limitation relative to literature datasets noted by prior work on recourse in competitive settings (Fonseca et al., 2023)—it can still be used to sample candidates' initial features, grounding them in a realistic setting. Below, we describe how our experimental setup is adapted for this dataset and present preliminary results.

---

[1]https://archive.ics.uci.edu/dataset/522/south+german+credit

| | Time step | | | | | | | |
|---|---|---|---|---|---|---|---|---|
| | $t = 1$ | $t = 2$ | $t = 3$ | $t = 4$ | $t = 5$ | $t = 6$ | $t = 7$ | $t = 8$ |
| **Threshold values** | 0.46 | 0.63 | 0.60 | 0.63 | 0.56 | 0.57 | 0.59 | 0.54 |
| **Recommended scores** | 0.63 | 0.60 | 0.63 | 0.63 | 0.58 | 0.59 | 0.60 | |
| **Recourse Reliability** $RR_t$ | | 1.0 | 1.0 | 1.0 | 1.0 | 1.0 | 1.0 | 1.0 |
| **Recourse Feasibility** $RF_t$ | | 0.36 | 0.67 | 0.30 | 0.86 | 0.57 | 0.71 | 0.75 |

Table 6: Values of the threshold, the subsequent score recommended by the predictor $\mu$, and the observed Recourse Reliability ($RR_t$) and Recourse Feasibility ($RF_t$) at each time step in an 8-step evaluation episode. The quantities $RR_t$ and $RF_t$ are computed relative to the recommendation issued in the previous step. At $t = 2$ and $t = 4$, the agent recommends a target score lower than the most recently observed threshold. The corresponding values of $RR_t$ and $RF_t$ at the following steps ($t = 3$ and $t = 5$) are both desirable.

| | Time step | | | | | | | |
|---|---|---|---|---|---|---|---|---|
| | $t = 1$ | $t = 2$ | $t = 3$ | $t = 4$ | $t = 5$ | $t = 6$ | $t = 7$ | $t = 8$ |
| **Threshold values** | 0.55 | 0.57 | 0.59 | 0.60 | 0.58 | 0.62 | 0.55 | 0.61 |
| **Recommended scores** | 0.61 | 0.63 | 0.65 | 0.66 | 0.64 | 0.68 | 0.61 | |
| **Recourse Reliability** $RR_t$ | | 1.0 | 1.0 | 1.0 | 1.0 | 1.0 | 1.0 | 1.0 |
| **Recourse Feasibility** $RF_t$ | | 0.36 | 0.44 | 0.33 | 0.71 | 0.71 | 0.43 | 0.50 |

Table 7: Values of the threshold, the subsequent score recommended by ARR, and the observed Recourse Reliability ($RR_t$) and Recourse Feasibility ($RF_t$) at each time step in an 8-step evaluation episode. The quantities $RR_t$ and $RF_t$ are computed relative to the recommendation issued in the previous step.

### G.1 Experimental set-up

We use the German Credit dataset to train the predictive model $M(\cdot)$ and to sample candidates' initial features. Since the dataset only contains $1\,000$ samples, which is insufficient to both train $M(\cdot)$ and generate a diverse set of candidate profiles for all time steps across episodes, we augment it with synthetic data, generated to be as similar as possible to the original samples.

To preserve the statistical properties of the original dataset, we employ a Gaussian copula approach (Nelsen, 2006), that captures both the marginal distributions and the correlation structure among features, thus ensuring that synthetic samples closely resemble the original data in terms of both individual feature distributions and inter-feature dependencies.

A key difference between the German dataset and the synthetic dataset that we use in our main experiments is the presence of categorical features. This changes the meaning of feature difficulties: while for continuous features they represent a relative difficulty to implement a feature change, in the case of categorical features they represent the difficulty of switching from a category to another. Thus, for every categorical feature we define a matrix of difficulty parameters, one for every ordered pair of categories. These parameters vary both within the same category (changing the savings from $< 100$ to $\geq 1000$ is more difficult than changing it to $100 \leq x < 1000$), and among different categories (changing the purpose of the loan is easier than changing the savings or the employment). These values represent the difficulty of implementing the corresponding category switch.

The probability of successfully implementing a categorical feature switch $p_{\text{success}}$ is also different from Equation 8, due to these changes. Specifically, we design it as:

$$p_{\text{success}}^{\text{cat}} = (1 - d_{i,j})^{\frac{1}{\beta_{\text{cat}}}} \tag{19}$$

where $\beta_{\text{cat}}$ is a setting difficulty parameter, analogous to the parameter $\beta$ previously defined. Lower values of $\beta_{\text{cat}}$ indicate settings where implementing changes to categorical features is generally more difficult. Instead, $d_{i,j}$ represents the difficulty of switching from category $i$ to category $j$.

### G.2 Implementation details

We train our recommender agent in an environment similar to the one described in Section 3.1 and Appendix B, with two key modifications: (i) recommendations on categorical features are constrained to category switches, and (ii) the probability of successfully implementing a category switch follows the formulation defined in Equation 19.

The recommender requires a longer training period compared to the synthetic dataset experiments. This increased complexity likely stems from the challenges inherent in categorical feature recommendations. Specifically, with 17 categorical features spanning 54 distinct categories, the agent must navigate a substantially larger action space of possible feature switches. Moreover, category switches induce non-smooth changes in the credit score, making it difficult to identify counterfactuals that achieve a precise target score. To account for this complexity, we train the recommender for $20\,000$ time-steps in the first phase and $30\,000$ time-steps in the second phase.

The predictor, as in the experiments on synthetic samples, is trained for $7\,000$ time-steps. We use the following hyperparameters: $T = 1$, $\beta = 0.05$ and $\beta_{\text{cat}} = 1.0$.

### G.3 Results

In Table 8, we report the results of the preliminary experiment on the German dataset. As can be seen, both our method and the hybrid variant achieve desirable levels of reliability in this more complex setting. On the other hand, as in the scenario with synthetic data, the baseline's threshold-based policy yields recommendations that too many candidates can satisfy, thereby reducing reliability.

These results showcase the utility of our approach in more complex scenarios, where feature values are drawn from a real dataset and both continuous and categorical features are taken into account.

| | Method | | |
|---|---|---|---|
| | **Baseline** | **Hybrid** | **Ours** |
| **Recourse Reliability** $RR_t$ | $0.43 \pm 0.05$ | $0.90 \pm 0.04$ | $0.89 \pm 0.03$ |
| **Recourse Feasibility** $RF_t$ | $0.78 \pm 0.06$ | $0.36 \pm 0.05$ | $0.40 \pm 0.03$ |

Table 8: Results for the baseline, the hybrid method (based on Ustun), and our approach in the simulation environment based on German, with $\beta = 0.05$, $\beta_{\text{cat}} = 1.0$, and $T = 1$.

## H  DISCUSSION ON CONVERGENCE GUARANTEES

NEW

While our method and the hybrid variants yield desirable solutions in all four settings under consideration ($T \in \{1, 5\}$, $\beta \in \{0.05, 0.01\}$), we cannot provide theoretical guarantees that the agent converges to an *optimal* solution. This limitation arises because the reinforcement learning algorithm we employ, Soft Actor Critic (SAC) (Haarnoja et al., 2018), does not include such guarantees.

In their paper, the authors provide lemmas and theorems establishing convergence of Soft Policy Iteration, the tabular algorithm SAC is based on. However, these results rely on assumptions that SAC does not satisfy. SAC uses neural networks to approximate policies and value functions, which breaks the assumption of exact or tabular representations, and it operates in continuous action spaces. Moreover, SAC incorporates entropy maximization and stochastic actor updates, which further depart from the theoretical setting in which convergence can be proven.

At the same time, deriving strong theoretical bounds in our environment would be extremely difficult, due to the complexity of the environment. Candidates' behavior is noisy, the system is only partially observable, and the dynamics involve significant stochasticity. Given these complexities, this work is focused on establishing a rigorous empirical framework that demonstrates the practical effectiveness of our approach across diverse experimental conditions.

## I  USE OF LARGE LANGUAGE MODELS

Large language models were used solely to improve the clarity and grammar of the text and to generate the icons of candidates in Figure 1. All substantive content was written by the authors; LLMs were applied only for minor phrasing refinements.

