# OpenReview forum: "Reinforcement Learning for Durable Algorithmic Recourse"
_ICLR.cc/2026/Conference — Submitted to ICLR 2026_

### Official Review · Reviewer_SXeV · 2025-10-24

**Soundness:** 2
**Presentation:** 3
**Contribution:** 2
**Rating:** 4
**Confidence:** 4

**Summary:**

This paper introduces a new framework of algorithmic recourse to address the challenge of providing durable and effective recourse recommendations.
The authors focus on competitive resource-constrained environments (e.g., admissions, loan approvals), where providing recourse recommendations to rejected applicants can alter the applicant pool and shift the decision boundary.
The authors propose a concrete solution based on reinforcement learning that accounts for the dynamics of the environment to generate feasible and valid recourse recommendations.
By experiments on synthetic data, the proposed method was evaluated against existing standard algorithmic recourse methods.

**Strengths:**

1. This paper is well-written, well-organized, and easy to follow.
1. The authors tackle an underexplored problem in the field of algorithmic recourse. I agree with the importance of ensuring long-term validity in dynamic environments. The authors also propose a first concrete solution to this problem. I think it is interesting to leverage reinforcement learning to model dynamics and optimize recourse recommendations that are both valid and feasible.

**Weaknesses:**

1. One of my main concerns is the lack of theoretical analyses. For example, there is no theoretical guarantee of the cumulative reward achieved by the proposed algorithm. While I acknowledge that this paper focuses on establishing a foundation for the proposed problem, I think providing at least a preliminary theoretical discussion or bounds, even under simplified assumptions, would significantly strengthen the paper. Such an analysis could help clarify the conditions under which the proposed method is expected to perform well.
1. Another concern is the simplified experimental setting. For example, all features are continuous and drawn from independent distributions. In addition, the authors employed a logistic regression model as a prediction model $M$. However, real-world scenarios involve correlated or categorical features, as well as more complex decision boundaries produced by major models, such as neural networks or tree ensembles. Thus, the current experimental setting seems too simple to validate the efficacy of the proposed method.
1. I am also concerned that the proposed method appears to require a substantial amount of interaction to learn an effective policy. In Appendix, the authors note that training involves tens of thousands of episodes in simulation. In many high-stakes decision-making tasks, such a large number of interactions may be costly and impractical. Thus, I am concerned that the proposed method may not work well in more practical situations.

**Questions:**

1. Some previous studies, such as (Wu et al. 2024) and (Kanamori et al. 2025), also consider algorithmic recourse in the online setting and conducted experiments on real datasets. As done in these studies, could the authors also conduct simulation experiments on real datasets?

---

> ### Author Response · Authors · 2025-11-19
>
> **Weakness 1: One of my main concerns is the lack of theoretical analyses. For example, there is no theoretical guarantee of the cumulative reward achieved by the proposed algorithm. While I acknowledge that this paper focuses on establishing a foundation for the proposed problem, I think providing at least a preliminary theoretical discussion or bounds, even under simplified assumptions, would significantly strengthen the paper. Such an analysis could help clarify the conditions under which the proposed method is expected to perform well.**
>
> We thank the reviewer for bringing up the question of theoretical analyses. Unfortunately, we are not able to provide such theoretical bounds, as the chosen reinforcement learning method (Soft Actor Critic) lacks guarantees. Indeed, in the original paper introducing Soft Actor Critic (SAC) (Haarnoja et al., 2018), the authors present lemmas and theorems on the convergence to an optimal policy of Soft Policy Iteration, the method that SAC is based on. However, such guarantees rely on assumptions that SAC breaks. Specifically, SAC uses neural networks to learn the policy and value functions, hence breaking the assumption of tabular or exact representations, and operates in continuous action spaces.
>
> As a result, we cannot provide convergence guarantees. We recognize that the lack of such analysis is an important limitation. At the same time, we think deriving strong theoretical bounds in a system with significant stochasticity, as candidates’ behavior is noisy, and partial observability, would be very difficult. Instead, we have focused on creating a solid framework, through an empirical design, and showing the utility of our proposed solution across various dimensions. If the reviewer finds it useful, we are open to explaining SAC’s lack of theoretical bounds further and adding a dedicated discussion to the manuscript.
>
> **Weakness 2: Another concern is the simplified experimental setting. For example, all features are continuous and drawn from independent distributions. In addition, the authors employed a logistic regression model as a prediction model. However, real-world scenarios involve correlated or categorical features, as well as more complex decision boundaries produced by major models, such as neural networks or tree ensembles. Thus, the current experimental setting seems too simple to validate the efficacy of the proposed method.**
>
> Thank you for this observation. In response to feedback from multiple reviewers regarding real-world validation, we are conducting additional experiments on the German Credit dataset. This dataset contains both continuous and categorical features, hence its addition should also respond to some of your concerns. In this new setting, we aim to use an MLP as predictor, to capture more complex decision boundaries. We hope to complete these experiments before the end of the discussion period and will provide an update as soon as results are available.
>
> We note that our choice of a simplified setting was motivated by addressing the open problem introduced by Fonseca et al. (2023), who first identified the challenge of recourse invalidity in competitive environments. We initially designed our experiments to build upon their simulation framework. While we extended it to incorporate more complex dynamics, we maintained some simplifications to establish a foundational solution to this previously unsolved problem. That said, we acknowledge that evaluation on real-world datasets could further strengthen our work, and we hope to deliver robust results in this setting as well.

---

> ### Author Response · Authors · 2025-11-19
>
> **Weakness 3: I am also concerned that the proposed method appears to require a substantial amount of interaction to learn an effective policy. In Appendix, the authors note that training involves tens of thousands of episodes in simulation. In many high-stakes decision-making tasks, such a large number of interactions may be costly and impractical. Thus, I am concerned that the proposed method may not work well in more practical situations.**
>
> Thank you for raising this concern. Reviewer cToo also brought up this issue, and we agree that it’s an important point that needs a more in-depth discussion. For this reason, we have extended this point in the Discussion and Conclusion section.
>
> While it is true that several training episodes are required for convergence, this does not imply that all training must be performed on real applicants, nor that performance on early candidates must be sacrificed. There are well-established ways to mitigate this issue.
>
> A natural solution, as mentioned in the paper, is to shift to offline reinforcement learning, which trains policies directly from previous interactions rather than through online exploration. This procedure requires data on past applicants that received recourse and acted on it (or decided not to reapply). If a recourse process (even a naive one) is already in place, recourse providers could share this historical data. This would allow us to pre-train both the recommender and the predictor using offline RL techniques, which are specifically designed to learn robust policies from past data without unsafe exploration. An important characteristic that such data would require for optimal convergence of our agents is coverage; hence, it should encapsulate a variety of situations so that the agents learn to generalize.
>
> If data on past applicants was not available, we could use the simulation environment to pre-train the agent. In this case, domain knowledge could guide reasonable assumptions about the unknown factors: for example, the overall difficulty of the setting or the relative difficulty of modifying each feature. When the system is subsequently deployed and begins receiving real online data, the initial policy may not be optimal, but it should still outperform an entirely untrained agent. This would ensure that performance on early (transient) candidates is not fully sacrificed, while also allowing the agent to adapt and converge more quickly in the real environment.
>
> In summary, although the lack of real-world data and the need for safe policies are well-known challenges in RL, leveraging a sufficiently complex simulator and domain knowledge would enable us to mitigate these limitations.
>
> **Question 1: Some previous studies, such as (Wu et al. 2024) and (Kanamori et al. 2025), also consider algorithmic recourse in the online setting and conducted experiments on real datasets. As done in these studies, could the authors also conduct simulation experiments on real datasets?**
>
> Thank you for raising this important point. As mentioned above, in response to this comment and other reviewers’ comments, we have started an implementation on the German Credit dataset and hope to provide numerical results soon.
>
> As for the reviewer’s questions, our framework requires modelling how individuals’ features evolve over multiple application rounds. Because real datasets (e.g., German, Adult) contain only a single snapshot per individual and no information on if, when and how candidates reapply, we needed to synthetically simulate the evolution environment. Given that the temporal dynamics had to be simulated, we initially chose to simulate the initial states as well, rather than grounding them in a specific dataset, following prior work’s procedure (Fonseca et al., 2023, Bell et al., 2024). That said, we recognize that initializing candidate features from real data could further strengthen our work.
>
> The papers mentioned by the reviewer follow a similar overall approach:
>
> - Wu et al. (2024) also use synthetic transition dynamics, but they initialize trajectories from real data points (e.g., Adult, German Credit).
>
> - Kanamori et al. (2025) likewise use real datasets only as starting points, while the improvement process itself is modelled synthetically through an online bandit formulation.

---

> ### Comment · Reviewer_SXeV · 2025-11-28
>
> Thank you for taking the time to answer all my questions and address my concerns. My concerns were partially addressed in the rebuttal. After reading the other reviews, I have decided to increase my score. While the lack of theoretical guarantees remains a key weakness in my view, I now understand the difficulty of providing such analysis after the rebuttal. I hope the final version will include the discussion on Weakness 1, as well as the experimental results on the German dataset.

---

> > ### Author Response · Authors · 2025-11-29
> >
> > Thank you for your feedback. We have now added a discussion on convergence guarantees in Appendix H, and mentioned this limitation in the Discussion and Conclusions section. Additionally, the experimental setup for our experiment on German and the results on this dataset, which are aligned to the results on the synthetic dataset, are described in Appendix G.

---

### Official Review · Reviewer_3gTN · 2025-10-30

**Soundness:** 3
**Presentation:** 2
**Contribution:** 2
**Rating:** 4
**Confidence:** 4

**Summary:**

The paper studies the problem of ensuring robust recourse suggestions in dynamic and resource-constrained environments. The authors first characterize a novel simulation environment, and then provide a reinforcement learning solution to provide population recourse suggestions that stay valid for a longer period of time. Lastly, the authors validate their approach in synthetic experiments comparing their approach with the state of the art.

**Strengths:**

The paper’s topic is interesting and original. It departs from the classical literature and it considers a more realistic environment where users also compete for resources. The simulation environment is interesting, and it provides additional options to simulate complex scenarios for recourse. If extended and/or tailored with some causal flavor (e.g., [1]), I believe it could become a nice synthetic benchmark for recourse algorithms and/or in general social decision making.

[1] Cinquini, Martina, et al. "A Practical Approach to Causal Inference over Time." Proceedings of the AAAI Conference on Artificial Intelligence. Vol. 39. No. 14. 2025.

**Weaknesses:**

I believe Section 4 is not very clear in its exposition. The proposed setup is somewhat non-standard for recourse, and some of the assumptions they make are not very detailed (e.g., access to the POMDP, stochasticity of the bank’s score $g$).

- The recourse policy $\phi$ and the predictor policy $\mu$ are trained with two different environments (as specified by 4.1 and 4.2). It is not clear the reason for this choice. Indeed, having access to historical traces, or directly the POMDP environment, is already a very strong assumption, so I would have expected to have also $\phi$ trained with it.
- The paper does not compare with the “naive” baseline, which is robustifying recourse with a simple $\epsilon$-ball, as is done in [2]. Indeed, the proposed approach requires building a very expensive world model (Section 4), while simple robustification solutions could be more viable in practice. As shown by [3], robustifying recourse with an $\epsilon$-ball can still provide some robustness to time. Although both [2,3] consider a causal setting, I believe a comparison with [2] could have been contemplated, or at least a discussion about why an RL approach is better.
- The simulation environment (Section 3.1) feels very abstract. Although it extends previous results, the authors do not describe potential real-world phenomena or population trends they could simulate with it. For example, they could have considered more challenging or interesting scenarios (e.g., different groups with different urgency/self-confidence that compete) to show how recourse changes the system dynamics.
- The notion of “equity” in the reward function (Equation 1) should be better contextualized. Indeed, the classifier score could be an imperfect proxy of the difficulty of a task. Thus, we might end up providing similar actions to diverse participants that produce the same score, but which require a very different effort per user.

[2] Dominguez-Olmedo, Ricardo, Amir H. Karimi, and Bernhard Schölkopf. "On the adversarial robustness of causal algorithmic recourse." International Conference on Machine Learning. PMLR, 2022.

[3] De Toni, Giovanni, et al. "Time can invalidate algorithmic recourse." Proceedings of the 2025 ACM Conference on Fairness, Accountability, and Transparency. 2025.

**Questions:**

Here, I leave some questions for the authors. The “mandatory” ones (I apologize for the lack of a better term) are those that could entail an improvement of the score if answered satisfactorily.

**[Mandatory Questions]**
- How does your approach compare with a robustified recourse with an $\epsilon$-ball as in [2] in your scenario? Alternatively, what are the advantages of your approach with respect to [2]?

**[Optional Questions]**
- Could you comment on the notion of “equity” you describe in your paper (Equation 1)?
- Can you comment on potentially more complex scenarios you could simulate with your environment (Section 3.1)?

---

> ### Author Response · Authors · 2025-11-19
>
> **Weakness 1: I believe Section 4 is not very clear in its exposition. The proposed setup is somewhat non-standard for recourse, and some of the assumptions they make are not very detailed (e.g., access to the POMDP, stochasticity of the bank’s score).**
>
> **- The recourse policy and the predictor policy are trained with two different environments (as specified by 4.1 and 4.2). It is not clear the reason for this choice. Indeed, having access to historical traces, or directly the POMDP environment, is already a very strong assumption, so I would have expected to have also trained with it.**
>
> Thank you for pointing this out; we do realize that Section 4 introduces important concepts for our method and that it needed refinement to improve its clarity. On this note, we are not entirely sure what the reviewer means by “access to the POMDP”. We would like to note that the agent does not have access to the POMDP in the sense of information on the environment. The POMDP is our formalization of the simulation environment, and the agent, representing the recourse system, interacts with it by observing applying candidates and providing them with recommendations, as any real recourse system would. While we who design the simulated environment know details about it, such as the hyperparameters, the agent is not given any additional information about the environment or about the underlying POMDP that would not be available to the recourse system in a real-world setting. Similarly, we do not view access to historical traces as a strong assumption, as these traces correspond to candidate characteristics that the agent has observed at some earlier point in time. However, we are open to different opinions on this matter and to clarifying this in the manuscript.
>
> Regarding the stochasticity of the goal score $g$, this value is sampled randomly during training so that the agent learns to provide recommendations that push candidates toward a variety of possible goal scores. This variability helps the agent generalize and learn more robustly.
>
> The reason the predictor and recommender are trained in two different environments is that, although they operate within the same overall recourse setting, they have fundamentally different objectives. The predictor’s goal is to learn how candidates adapt to recommendations and to predict future competition, and therefore it requires access to the full environment dynamics. In contrast, the recommender aims to learn feature difficulties and to identify the optimal recommendation path for any given feature vector and goal score. The environment used for training the recommender can therefore be seen as a simplification of the full environment (used for training the predictor): it focuses on a single candidate, provides less information (since additional details are unnecessary for its task), and uses a reward function solely centered on the suitability of the recommended feature changes.
>
> We have incorporated some of these clarifications in Section 4.
>
> **Weakness 2: The paper does not compare with the “naive” baseline, which is robustifying recourse with a simple $\epsilon$-ball, as is done in [2]. Indeed, the proposed approach requires building a very expensive world model (Section 4), while simple robustification solutions could be more viable in practice. As shown by [3], robustifying recourse with an $\epsilon$-ball can still provide some robustness to time. Although both [2,3] consider a causal setting, I believe a comparison with [2] could have been contemplated, or at least a discussion about why an RL approach is better.**
>
> **Question 1: How does your approach compare with a robustified recourse with an $\epsilon$-ball as in [2] in your scenario? Alternatively, what are the advantages of your approach with respect to [2]?**
>
> Thank you for your comment. We plan to implement Dominiguez-Olmedo et al.’s approach [2], and we hope to provide some results before the end of the discussion period.
>
> Regarding the comparison with the $\epsilon$-ball robustification used in [2], we note that while such robustification may be effective in scenarios where there are truly unpredictable changes against which one aims to be robust (exogenous changes to the prediction model), in our scenario the behavior of candidates, although noisy, can be learned. This makes it possible to predict shifts in competition rather than robustify blindly. As a result, the recourse system avoids robustifying when the competition is not expected to increase and avoids over-robustifying when it is not needed.
>
> This suggests that our approach should be more suitable than Dominiguez-Olmedo et al.’s method in this setting, although we hope to support this expectation with numerical results.

---

> ### Author Response · Authors · 2025-11-19
>
> **Weakness 3: The simulation environment (Section 3.1) feels very abstract. Although it extends previous results, the authors do not describe potential real-world phenomena or population trends they could simulate with it. For example, they could have considered more challenging or interesting scenarios (e.g., different groups with different urgency/self-confidence that compete) to show how recourse changes the system dynamics.**
>
> **Question 3: Can you comment on potentially more complex scenarios you could simulate with your environment (Section 3.1)?**
>
> We agree with the reviewer that more complex and noisy simulations could be valuable extensions for future work. Our framework is designed to be extensible, and personalization phenomena such as those mentioned by the reviewer (different groups with different urgency/self-confidence) could easily be added to enhance complexity. Other interesting phenomena to incorporate include exogenous changes to the prediction model and external population, collective action of candidates, and users with a priori low trust in the recourse system who decide not to engage with it.
>
> In the current simulation, we focus on modeling how candidates react to recommendations: whether and how they implement them, and whether and when they choose to reapply. This behavior is governed by stochastic probability functions that incorporate recourse-related factors, capturing several complex, and in our opinion realistic, phenomena, such as candidates dropping out after multiple unsuccessful applications despite getting closer to acceptance, or candidates struggling with difficult feature changes and therefore taking more time. While all candidates currently follow the same probability functions (a simplification we acknowledge), the stochastic nature introduces heterogeneity in candidate behavior, with some candidates behaving in unexpected or atypical ways.
>
> We have expanded Section 3.1 to provide a more detailed description of the simulation environment. If the reviewer finds it useful, we can also extend Appendix B to discuss how the environment could be further adapted to incorporate more complex phenomena, such as those described above.
>
> **Weakness 4: The notion of “equity” in the reward function (Equation 1) should be better contextualized. Indeed, the classifier score could be an imperfect proxy of the difficulty of a task. Thus, we might end up providing similar actions to diverse participants that produce the same score, but which require a very different effort per user.**
>
> **Question 2: Could you comment on the notion of “equity” you describe in your paper (Equation 1)?**
>
> Thank you for bringing up this point. We agree that it needed a more in-depth discussion in our manuscript, which we have now added.
>
> The notion of equity in our reward function (Equation 1) comes from the idea that all candidates should be given the same score target. The reviewer is correct that achieving the same score may entail different levels of effort for different candidates. While effort is an aspect that one may decide to focus on—ensuring that different recommendations entail similar effort—we adopt an equity view based on merit, represented in this case by the model score. In other words, we ensure similar effort for people with similar merit and different effort for people with different merit, scaled by the difference in merit.
>
> The reason for this choice is that when the recourse system recommends a goal score, the system is essentially telling candidates that this goal score will be enough to get accepted. If we did not enforce this criterion, we would be telling two different candidates that they need two different levels of qualification to be accepted by our system. We envision that such a scenario would not be appreciated by candidates who get recommended higher scores.
>
> From a computational perspective, relaxing this equity constraint would also allow the agent to learn very unbalanced policies. Indeed, to maximize the reward, it could learn to “sacrifice” a subset of candidates by giving them extremely high recommendation scores that they are unlikely to achieve, while assigning very low targets to the remaining candidates. The latter group would then be accepted almost automatically once the others quit. Although this would result in high reliability, and infeasibility only for the “sacrificed” candidates, it would represent an unfair and highly undesirable outcome.

---

> ### Comment · Reviewer_3gTN · 2025-11-27
>
> I thank the authors for the detailed response. Please find below some additional comments/suggestions.
>
> **[Learning environment]** What I meant is that, in order to train both the recommender and the predictor policy, you need to have some sort of “simulator” that generates traces similarly to the underlying environment. In practice, that might be as hard to achieve as to learn the recourse policies. Further, regarding the historical traces, I believe the hardness stems from their “granularity”. For example, how many data points for the same user could I observe in a $[0,t]$ timeframe? However, this is a common issue of recourse approaches, shared by many works, so for now we have to live with it.
>
> **[$\epsilon$-ball recourse]** I fear that, in practice, an $\epsilon$-ball robustification would be more viable than having to learn the entire user’s behaviour from data, which is why I asked for a simple comparison between the approaches. Nevertheless, I understand the differences between the proposed approach and Dominiguez-Olmedo et al., so I guess that, having access to some reasonable population data (as an example [1]), one would favour more targeted robustification.
>
> I look forward to seeing some (preliminary) results in that sense.
>
> **[Section 3.1 Phenomena]** I believe some sort of description of potential further scenarios your simulation environment could support would strengthen the paper and foster further contributions on top of it. I suggest that the authors add some discussion in that sense (in the Appendix would be more than fine).
>
> After reading the other reviews, I have updated my score accordingly (hoping to see some additional results before the discussion periods end).
>
> [1] https://data.census.gov/profile

---

> ### Author Response · Authors · 2025-11-29
>
> Thank you for your thoughtful response; below we address your additional comments.
>
> **[Learning environment]**  We now understand what you mean when referring to access to the MDP, and agree that our simulator relies on assumptions about the environment and parameters that are not precisely available in practice. On this matter, we wish to point out that, in practice, a perfect simulator would likely not be needed to train our agent. Indeed, as noted in our responses to reviewers cToo and SXeV, the agent could be *pre-trained* in our simulation environment, where the parameters modelling the environment would be chosen by domain experts to closely reflect its characteristics (e.g., overall difficulty of the setting, feature difficulties). This pre-trained agent could then be trained online on real candidates who apply for recourse in the setting where we aim to deploy our algorithm.
>
> We acknowledge that, at the beginning of training on real candidates, imperfect alignment between the simulator and the real environment may lead to suboptimal recommendations; however, we believe that such pre-training would still accelerate convergence. As mentioned in other responses, offline training could also be a viable solution. To summarize, since no data regarding recourse environments and the behavior of candidates in such environments is currently available, we have opted to simulate it; however, in a practical application, real data on candidates’ behavior would be observed and used to ensure that the agent learns the dynamics of the true environment.
>
> **[$\varepsilon$-ball recourse]** Thank you for pointing this out. We have implemented a comparison with an $\varepsilon$-ball approach and added the results to our manuscript in Section 5. You are right that such an approach is simpler, as it does not require training or learning the environment’s dynamics. However, as noted in the manuscript, while this naïve approach achieves similar results to our hybrid strategy (combining Ustun’s method with our predictor) when reliability requirements are not stringent, in high-reliability regimes our hybrid approach exhibits Pareto optimality. This is because our predictor does not always robustify: there are time steps where the agent predicts a decrease in competition and thus recommends a score lower than the previous threshold, while still guaranteeing very high reliability and thereby achieving higher feasibility. In contrast, the $\varepsilon$-ball approach always suggests target scores that exceed the current threshold. This in-depth analysis has been added to Appendix F.
>
> In other words, having access to data on how candidates behave in response to recourse allows for a grounded robustification, that achieves a better trade-off with respect to the naïve approach. Moreover, in Appendix G, we show that our method is effective even on data coming from a real-world source, in this case the German credit dataset.
>
> **[Section 3.1 Phenomena]** Thank you for your suggestion, which we agree can help to motivate extensions of our work. Accordingly, we have incorporated an expanded discussion in Appendix B.5, where we describe additional phenomena that could be simulated and detail how our environment formulations would need to be modified to support these extensions.

---

### Official Review · Reviewer_cToo · 2025-10-31

**Soundness:** 3
**Presentation:** 4
**Contribution:** 3
**Rating:** 8
**Confidence:** 4

**Summary:**

This paper proposes a reinforcement-learning (RL) framework for *durable algorithmic recourse*—recommendations that remain valid over time in competitive, resource-limited settings. The authors model the recourse process as a partially observable Markov decision process (POMDP), capturing how applicants’ responses influence future environments. A hierarchical RL design is introduced with two policies: a recourse recommender φ that generates counterfactual feature vectors for rejected individuals, and a predictor μ that selects target scores to balance equity, feasibility, and validity over a horizon T. The recommender adaptively learns feature-specific modification difficulties from observed success rates, while the predictor anticipates population dynamics to maintain reliability under competition. Experiments on synthetic settings show improved Pareto trade-offs between Recourse Reliability (RR_t^T) and Feasibility (RF_t^T) compared to threshold-based and optimization-based baselines such as Ustun, Wachter, and DiCE.

**Strengths:**

The work formalizes the new notion of *durable recourse*, explicitly modelling time-dependent validity in dynamic, competitive environments. It is, to my knowledge, the first to cast recourse generation as a hierarchical RL problem, addressing endogenous feedback effects. The reward structure is well designed, combining the Gini index for equity with validity and feasibility objectives. The two-level decomposition (recommender and predictor) is a thoughtful design choice that improves computational stability and interpretability. The presentation is clear, with detailed formalism, well-motivated objectives, and effective figures illustrating Pareto frontiers and convergence behaviours. The paper addresses an important gap in the literature by moving beyond static or exogenous-robust recourse, and it provides a principled direction for future work connecting RL, fairness, and causal inference.

**Weaknesses:**

The RL training procedure is computationally heavy: convergence requires around 23,000 episodes for the recommender and 7,000 for the predictor, corresponding to unrealistically long simulated cycles. Although offline training is mentioned, there is no empirical demonstration or discussion of the data requirements for such a setup. The trained policies remain static and cannot adapt to changing models or applicant distributions, which undermines the claim of robustness to environmental shifts. All experiments rely on synthetic datasets with small populations (N=20, d=10) and simplified normal feature distributions; real-world evaluation on credit or recidivism data would strengthen the paper’s impact. The use of a fixed T-step history as a proxy for partial observability is a heuristic; there is no comparison against recurrent or belief-state approaches, nor analysis of the sufficiency of T. Finally, the ethical framing, though present, could better connect the notion of durability with user trust and systemic accountability.

**Questions:**

1. The paper mentions offline RL as a possible direction. What data would be required to pre-train the policies before deployment, and how could such data be obtained without an existing recourse system?
2. How should practitioners select the time horizon T in real applications, where reapplication intervals vary among individuals?
3. Could the approach be extended with continual or meta-RL to adapt to distributional shifts or updated decision models?
4. Would results generalize to real-world datasets (e.g., COMPAS, FICO, or credit scoring)? Why or why not?
5. What are the computational and sample requirements for training both policies jointly, and how do they scale with feature dimensionality?

---

> ### Author Response · Authors · 2025-11-19
>
> **Weakness 1: The RL training procedure is computationally heavy: convergence requires around 23,000 episodes for the recommender and 7,000 for the predictor, corresponding to unrealistically long simulated cycles. Although offline training is mentioned, there is no empirical demonstration or discussion of the data requirements for such a setup.**
>
> **Question 1: The paper mentions offline RL as a possible direction. What data would be required to pre-train the policies before deployment, and how could such data be obtained without an existing recourse system?**
>
> We thank the reviewer for highlighting an important limitation that we had not discussed in sufficient depth. Our current experiments rely on simulated interactions with candidates, and convergence requires a relatively large number of episodes. The missing data in this setting concerns how candidates in a specific environment respond to recommendations, for example, how quickly they become discouraged, how easy it is for them to implement particular feature changes, and which features are easier to modify. In our simulation environment, these behaviors are encoded through parameters that model such factors. However, in a real-world deployment, the true values of these parameters are unknown, making it difficult to train the agent purely in simulation and then apply it directly to practice.
>
> A natural solution, as mentioned in the paper, is to shift to offline reinforcement learning, which trains policies directly from past interactions rather than through online exploration. If a recourse process (even a naive one) is already in place, recourse providers could share this historical data. This would allow us to pre-train both the recommender and the predictor using offline RL techniques, which are specifically designed to learn robust policies from previously collected trajectories without unsafe exploration. The main requirement for such training to be effective is sufficient data coverage: the dataset must contain a diverse range of situations so that the agent can generalize reliably.
>
> If data on past applicants were not available, we could use the simulation environment to pre-train the agent. In this case, domain knowledge could guide reasonable assumptions about the unknown factors, such as the overall difficulty of the setting or the relative difficulty of modifying each feature. When the system is subsequently trained on real online data, the initial policy may not be optimal, but it should still outperform an entirely untrained agent. This would ensure that performance on transient candidates is not fully sacrificed, while also allowing the agent to adapt and converge more quickly in the real environment. We have revised the Discussion and Conclusion section to incorporate a more in-depth discussion on this matter. If the reviewer finds it useful, we can further expand this matter in the Appendix.
>
> **Weakness 2: The trained policies remain static and cannot adapt to changing models or applicant distributions, which undermines the claim of robustness to environmental shifts.**
>
> **Question 3: Could the approach be extended with continual or meta-RL to adapt to distributional shifts or updated decision models?**
>
> Thank you for pointing this out. While our main focus is on endogenous shifts in the applicant pool, model shifts and shifts in the distribution of applicants are a known challenge and could be addressed in future work. Regarding model shifts, related concepts from the recourse literature can be particularly useful. For example, ROAR (Upadhyay et al., 2021) makes assumptions on worst-case model drift and provides recommendations that remain valid under such scenarios. A similar approach could be incorporated in our work: the predictor could continue to propose a target score that guarantees optimizes the reliability-feasibility trade-off under the current model, while the recommender accounts for model uncertainty and generates recommendations that would still achieve the target score even under worst-case shifts.
>
> Regarding shifts in the distribution of new candidates, we see a first promising direction in introducing such shifts already during training. In this way, the agent may learn to base its beliefs about the candidates’ distribution on recently observed applicants and thereby adapt its behavior to an evolving population even at test time. In addition, meta-RL approaches could be explored: by training the agent on a range of candidate distributions or model variations, meta-RL would allow the policy to adapt online to the specific shift encountered at test time. If the reviewer finds it useful, we can incorporate this discussion into the “Discussion and Conclusion” section, or alternatively in the Appendix.

---

> ### Author Response · Authors · 2025-11-19
>
> **Weakness 3: All experiments rely on synthetic datasets with small populations (N=20, d=10) and simplified normal feature distributions; real-world evaluation on credit or recidivism data would strengthen the paper’s impact.**
>
> **Question 4: Would results generalize to real-world datasets (e.g., COMPAS, FICO, or credit scoring)? Why or why not?**
>
> Thank you for raising this important point. Since multiple reviewers highlighted the lack of real-world evaluation as a limitation, we have begun running experiments on the German Credit dataset. We expect to have results before the end of the discussion period and will share them as soon as they are available.
>
> Regarding why we did not initially rely on real-world datasets: our framework requires information about how individuals’ features evolve over time across multiple application rounds. Standard real-world credit or recidivism datasets (e.g., German, COMPAS, FICO) only provide a single snapshot per individual rather than evolving data capturing repeated applications. This issue was already identified by Fonseca et al., which motivated the simulation environment they introduced and which we further refine in this work. As a result, even when a real dataset provides the initial feature values, we must still model the temporal evolution of those features through our simulation environment, informed by assumptions about feature difficulty and domain knowledge.
>
> We also note that the synthetic dataset used for training the prediction model is made up of 10,000 instances, and that, while the initial population of candidates at each episode is made up of only 20 candidates, 10 new candidates enter at each time step.
>
> We selected the German Credit dataset for our additional analysis because it closely aligns with our setting: loan approvals can involve competition under regulatory constraints and portfolio risk limits [1]. By contrast, our framework is not well-suited to recidivism prediction tasks such as COMPAS, where decisions about one defendant typically do not create competitive dynamics or directly constrain decisions on other defendants.
>
> **Weakness 4: The use of a fixed T-step history as a proxy for partial observability is a heuristic; there is no comparison against recurrent or belief-state approaches, nor analysis of the sufficiency of T.**
>
> **Question 2: How should practitioners select the time horizon T in real applications, where reapplication intervals vary among individuals?**
>
> Thank you for highlighting this point. Our use of a fixed $T$-length duration of validity is motivated by the observation that in many real applications, implementing recommended changes requires time. For example, in the German Credit setting, influential features such as job experience or income evolve slowly. A recommendation that assumes immediate reapplication with modified features would therefore be unrealistic and of limited practical value.
>
> In real deployments, practitioners could select $T$ by combining domain knowledge with empirical evidence. Domain expertise can help estimate the typical time required for individuals to implement meaningful changes to key features, allowing stakeholders to choose a horizon that balances recommendation validity with practical feasibility. Additionally, if historical data were available on applicants who received recommendations and later reapplied, one could empirically estimate the distribution of reapplication intervals or the time needed to implement specific changes. Such data would enable a more principled, data-driven choice of $T$. We have included part of this reasoning in Appendix A.
>
> We note that, as mentioned by the reviewer, $T$ also corresponds to the time window for which historical data is provided to the agent. This relies on the assumption that *candidates who reapply at time step $t$ were last rejected at some point in the window $[t-T, t-1]$*. This assumption follows from the fact that candidates are informed of the recommendation’s duration $T$, meaning they are essentially told: “If you implement these changes within the next $T$ time steps, you will be accepted.” Consequently, we believe it is not a strong assumption that all candidates who decide to reapply will do so within the next $T$ time steps: even if they have not perfectly implemented the recommendation by then, it is still convenient to reapply and receive a new recommendation, since the previous one is no longer going to be valid. Therefore, all the information on past candidates that the agent needs, in order to infer who is likely to reapply at the current time step and how, comes from the window $[t-T, t-1]$.
>
> [1] Mallett, Jacky. “What Are the Limits on Commercial Bank Lending?” Advances in Complex Systems, vol. 15, no. supp02, 2012, p. 1250075. DOI: 10.1142/S0219525912500750.
> URL https://www.worldscientific.com/doi/abs/10.1142/S0219525912500750

---

> ### Author Response · Authors · 2025-11-19
>
> **Question 5: What are the computational and sample requirements for training both policies jointly, and how do they scale with feature dimensionality?**
>
> Joint training of both policies is an interesting direction, but it is not one we have explored in the present work. We agree that simultaneous training could potentially accelerate convergence.
>
> In our current hierarchical approach, the sample and computational requirements of 23,000 and 7,000 episodes were determined empirically. We selected the number of training episodes for each policy through iterative experimentation, tuning until convergence was consistently achieved. As such, we do not yet provide theoretical guarantees on sample complexity.
>
> Regarding scalability with respect to feature dimensionality, our forthcoming experiments on the German Credit dataset will provide useful empirical evidence. After one-hot encoding of categorical variables, this dataset contains 54 features, significantly more than the synthetic settings. Training on this higher-dimensional space will allow us to observe how computational cost and convergence behavior scale in practice. We will report these results during the discussion phase as soon as they are available.
>
> **Weakness 5: Finally, the ethical framing, though present, could better connect the notion of durability with user trust and systemic accountability.**
>
> Thank you for pointing this out. As noted previously, our choice to make predictions robust over several time steps arises from the observation that, in many practical settings, the changes required for acceptance inevitably take time. In such contexts, providing users with recommendations that remain valid for longer periods gives those recommendations genuine actionability. This is ethically significant: as Venkatasubramanian et al. argue, the power of recourse, and thus the ability to alter one’s fate, is a modally robust good, and individuals should be provided with tools that allow them to meaningfully modify their outcomes. Offering candidates recommendations that are only valid for a single time step risks creating a false sense of agency, since they may have no realistic way to implement the required changes within that interval.
>
> Although the responsibility to ensure longer durability may not be as strong as the responsibility to ensure validity (since the latter corresponds more directly to breaking a promise), both are tied to user trust and to the system’s broader accountability. Failures on either front can lead candidates to disregard the recommendations altogether, ultimately rendering the recourse system ineffective. If the reviewer finds this clarification helpful, we are happy to incorporate this expanded ethical framing into the manuscript. We have extended Appendix A with this reasoning.

---

> ### Comment · Reviewer_cToo · 2025-11-26
>
> Thank you to the authors for the thoughtful and graceful responses. I appreciate the clarifications provided and the fact that several weaknesses are already being actively addressed, including new experiments on real data. I also recognize the inherent difficulty of evaluating a dynamic recourse algorithm in real-world settings, and I value the authors’ efforts to situate these challenges within the broader research landscape.
>
> In my view, the paper tackles an important and timely direction. The current draft makes promising contributions, and I trust the authors will continue strengthening the work by addressing the concerns raised—though I am aware that the extent to which further improvements can be incorporated before the camera-ready stage is ultimately constrained by the ICLR process.
>
> Having considered the other reviews, I still believe my original assessment reflects the significance and potential of this line of research. I would therefore like to maintain my score and trust the final decision to the area chair and the broader review process.

---

> > ### Author Response · Authors · 2025-11-27
> >
> > Thank you again for your thoughtful feedback and response, and for recognizing the challenges of the considered problem, and our contribution.

---

> > > ### Author Response · Authors · 2025-12-01
> > >
> > > We conducted an experiment using the German credit dataset. Appendix G details the experimental setup, including our data augmentation and implementation details. Our RL solution converged to a high-reliability configuration ($RR_t=0.89\pm0.03$, $RF_t=0.40\pm0.03$ ) after a slightly longer horizon due to a larger action space.

---

### Official Review · Reviewer_4FBv · 2025-11-01

**Soundness:** 4
**Presentation:** 3
**Contribution:** 2
**Rating:** 4
**Confidence:** 3

**Summary:**

The authors propose a RL approach to address algorithmic recourse in a competitive environment, where the action of each agent influence the chances of achieving recourse for the others.

**Strengths:**

Algorithmic recourse in competitive environments is a very sensible setting for real world applications, and very few works have started addressing it.

The RL method is carefully designed to maximise the realism of the setting.

**Weaknesses:**

Changes wrt baselines are not dramatic. Given that experiments are run in simulated settings, I am wondering whether there are conditions under which the proposed solution has larger advantages.

The novelty with respect to the works by Fonseca et al., 2023 and Bell et al., 2024 is not entirely clear. The authors do list assumptions being removed (end of section 3.1), but how much this affects the complexity of the task and the requirement for novel solutions is unclear.

**Questions:**

Can you better clarify the differences wrt existing solutions for competitive recourse? and their impact in the design of the method? isn't it possible to have experimental comparisons with their approach?

---

> ### Author Response · Authors · 2025-11-19
>
> **Weakness 1: Changes wrt baselines are not dramatic. Given that experiments are run in simulated settings, I am wondering whether there are conditions under which the proposed solution has larger advantages.**
>
> We interpret the reviewer’s comment on changes with respect to baselines as referring to the Pareto curves in Figure 2, where the improvements over the hybrid method appear less pronounced in the most stringent setting ($\beta = 0.01$). We thank the reviewer, and we agree that in this extreme regime the gains are modest. However, we would like to emphasize two points.
>
> First, the improvements become substantially more visible for $\beta = 0.05$, where our method achieves a clearer Pareto advantage (e.g., recourse reliability increases between 0.1 and 0.2 at the same feasibility level). Secondly, the hybrid method in this figure is not a literature baseline but rather a combination of a baseline with our own predictor; as such, it benefits from adaptability to the feasibility-reliability requirements of the system. In contrast, the true baseline produces a fixed solution (green cross), and under that comparison, our method yields a more notable improvement. In the revised version of the manuscript, we have changed the marker on the baseline in Figure 2 to underline that, unlike the hybrid method, this point corresponds to the true baseline taken from the literature.
>
> Regarding the reviewer’s question on conditions under which the proposed method shows larger advantages, our approach is particularly beneficial in scenarios that require high reliability. Although we demonstrate the adaptability of our approach to achieve various levels of reliability and feasibility, we emphasize that in real-world settings the former is typically more important, provided that a minimum level of feasibility is maintained. Indeed, stakeholders offering recourse expect recommendations to be generally valid; otherwise, candidates may lose trust in the system, rendering it ineffective. The baseline achieves very low reliability (∼40%), which means it would hardly be applicable in practice. Our method’s ability to adapt to strict reliability requirements provides a clear advantage when valid recourse is essential.
>
> Finally, the benefits relative to the hybrid method are strongest in settings such as $\beta = 0.05$ (i.e., when easy paths exist) and when implementing feature changes is not uniformly difficult across features. This applies to many real-world contexts, where the difficulty of modifying features varies substantially: for example, the German Credit dataset contains features related to property, job, and income (difficult), and others related to the duration of the loan or its purpose (easier). In such settings, the ability of our method to target features that candidates struggle with the least gives it an inherent advantage.

---

> ### Author Response · Authors · 2025-11-19
>
> **Weakness 2: The novelty with respect to the works by Fonseca et al., 2023 and Bell et al., 2024 is not entirely clear. The authors do list assumptions being removed (end of section 3.1), but how much this affects the complexity of the task and the requirement for novel solutions is unclear.**
>
> Thank you for the insightful comment. As noted, our work removes three key assumptions made by Fonseca et al. (2023) and Bell et al. (2024): (i) zero dropout probability (candidates leave only once accepted), (ii) uniform modification difficulty across features, and (iii) immediate reapplication. In other words, out of the three probabilities that we model (dropout probability, probability of success, and reapplication probability), Fonseca et al. and Bell et al. only modelled the second, hence the probability of successfully implementing the recommended changes. On the other hand, they treat the other two phenomena as deterministic: all candidates that have been rejected reapply, and they do so at every available time step. While they do mention that candidates may decide not to take action, in their framework, it means that they reapply (right away) with unchanged features.
>
> We introduce stochasticity by modelling both the probability of staying in the environment and the probability of reapplying. These probabilities depend on the recommendation received, as well as on other factors such as the number of previous applications. This added stochasticity increases the complexity of the framework, and therefore the agent’s learning problem, because outcomes of the agent’s actions are less deterministic. Nonetheless, we believe this leads to a substantially more realistic representation of real-life recourse, where such behavioral responses are inherently uncertain.
>
> For what concerns the probability of successfully implementing a recommendation, Fonseca et al. and Bell et al. model the probability of successfully implementing a recommendation based on the distance between the current score and the goal score, as well as a global difficulty parameter. We build upon this, observing that (1) rather than the magnitude of the recommended score change, the probability should depend on the magnitudes of recommended feature changes; (2) different features may have different inherent difficulties of being changed. As a result, we model one probability function per feature, representing the probability of implementing that specific feature change, which depends on an additional parameter called feature difficulty.
>
> Lastly, our framework involves an advancement in the synthetic data generation as well: Fonseca et al. and Bell et al. considered that candidates are generated over a 2-dimensional feature space, sampled independently at random. Each feature is sampled from a Gaussian distribution, where $x = (x_1, x_2)$ and $x_i \sim \mathcal{N}(\mu = 0.5, \sigma = 0.3)$, $i = 1, 2$. They mentioned training a simple logistic regressor as a classifier, and “The target variable $y_i$ for the prediction task is created randomly using a binomial distribution”.
>
> We improve this synthetic data generation by considering 10 continuous features instead of 2, and each feature is sampled from a different Gaussian. Moreover, we train the logistic regressor on a dataset where the features are constructed this way, and the ground truth target is constructed in order to be correlated with the features, with, of course, some noise. We refer to Appendix B.1 for this explanation.
>
> Regarding the reviewer’s question on whether these changes require novel solution approaches, we note that no previous solution has been proposed to the problem of recourse in competitive, limited-resource settings. Fonseca et al. and Bell et al. framed this problem as an open question. They proposed only a simulation setting, not a solution. We expanded their framework to account for higher uncertainty, which better reflects real-world situations, and proposed the first solution to this problem.
>
> We have expanded these points further in the manuscript, both in section 3.1 and appendix B.

---

> ### Author Response · Authors · 2025-11-19
>
> **Question 1: Can you better clarify the differences wrt existing solutions for competitive recourse? and their impact in the design of the method? isn't it possible to have experimental comparisons with their approach?**
>
> Thank you for the question. To the best of our knowledge, no existing method has been proposed to solve the competitive recourse problem in limited-resource settings. The prior works that study competitive recourse (Fonseca et al., 2023; Bell et al., 2024) explicitly identify it as an open problem and limit their contribution to introducing simulation environments rather than developing solutions. Their frameworks therefore do not include a method that could serve as a directly comparable baseline. This is why an experimental comparison with an existing approach, developed for the considered setting of competitive recourse, is, at present, not possible. We thank the reviewer for pointing out a potential lack of clarity in the research gap that we address. To make it clearer, we have revised the Related Works section, dividing it into small paragraphs related to: (i) Foundational works on recourse (focused mostly on static settings) (ii) Methods for dynamic recourse to address exogenous shifts (solutions for a different problem with respect to ours) (iii) RL methods for robust recourse in non-competitive settings (solutions for a different problem with respect to ours) (iv) Works by Fonseca et al. and Bell et al. that address the same problem as us, without proposing a solution.

---

> ### Comment · Reviewer_4FBv · 2025-11-25
>
> Thanks a lot for your detailed clarifications and updates, that allowed to better explain the novelty of the contribution wrt existing work. Of course how much the simulation scenario resembles a real-world situation is hard to say beyond the reasonableness of the design choices, but I am aware of the difficulty of evaluating such an algorithm in a real-life setting. It would however improve the realism of the framework if as many elements as possible (e.g. features) were drawn from existing real-world datasets. How complex is this given your setting?

---

> > ### Author Response · Authors · 2025-11-27
> >
> > Thank you for your feedback. We are working on an experiment using the German credit dataset and aim to provide results before the end of the discussion period. In this experiment, the prediction model $M(\cdot)$ is trained on the German dataset and the initial features of candidates in the environment are drawn from the dataset as well. German doesn't come with information on dynamics. How the candidates behave, i.e. whether/when they reapply after getting rejected, and whether they succeed in implementing feature changes, is governed by our simulation environment (i.e. dropout probability, probability of success, reapplication probability).

---

> > > ### Author Response · Authors · 2025-12-01
> > >
> > > We conducted an experiment using the German credit dataset. Appendix G details the experimental setup, including our data augmentation and implementation details. Our RL solution converged to a high-reliability configuration ($RR_t=0.89\pm0.03$, $RF_t=0.40\pm0.03$ ) after a slightly longer horizon due to a larger action space.

---

### Author Response · Authors · 2025-11-19

We thank the reviewers for their thoughtful comments, and for recognizing the novelty and impact of our solution in the considered setting, while highlighting important points that require deeper clarification. Specifically, the reviewers emphasize the importance of the considered problem, the originality of the solution, and the quality of the presentation. On the other hand, they underline the lack of testing on real data as a key limitation; moreover, some note that real-world applicability should be discussed in more depth, particularly in relation to the high number of training episodes required, and whether this limitation can be mitigated. Lastly, some reviewers point out that the simulation environment, and its advances with respect to the literature (Fonseca et al., 2023; Bell et al., 2024), should be discussed in more detail.

In response, we have provided clarifications in the rebuttal, and have made modifications to the manuscript to expand on aspects that were not discussed in sufficient depth. Additionally, we have begun an experiment on a real-world dataset (German Credit), and we aim to provide numerical results before the end of the discussion period. Lastly, in response to reviewer 3gTN’s request, we have begun investigating a comparison with a naive approach based on an $\epsilon$-ball robustification.

---

### Author Response · Authors · 2025-11-29

We thank all reviewers for their thoughtful feedback. A constructive rebuttal really contributed to our revised manuscript, where we have addressed the raised concerns. The major modifications are as follows:

- We added a comparison with a naïve approach based on an $\varepsilon$-ball, as requested by reviewer **3gTN**. The results of this experiment are reported in Section 5 and discussed in detail in Appendix F. Our method and the hybrid variant are Pareto-optimal with respect to this naïve approach ($RF_t = 0.71\pm0.01$ for our method and $RF_t = 0.57\pm0.02$ for the $\varepsilon$-ball approach, for $RR_t\approx0.95$).
- We included an experiment on the German Credit dataset to demonstrate the utility of our approach in real-life settings and when a larger number of features (including categorical ones) is considered, addressing the concerns of reviewer **SXeV** and **cToo**. These results are presented in Appendix G. Our method converges to a high-reliability configuration ($RR_t=0.89\pm0.03$, $RF_t=0.40\pm0.03$).
- We added a discussion explaining why theoretical convergence guarantees cannot be obtained in this setting, as requested by reviewer **SXeV**. This discussion appears in Appendix H and in Section 6.
- We expanded the discussion of the complex phenomena that can be simulated with our environment, as requested by reviewer **3gTN**. This has been added to Appendix B.5.
- We enhanced the Results section (Section 5) to more clearly distinguish between our method, the proposed hybrid method, and the baseline, and to emphasize that a high level of reliability is generally required. This addresses reviewer **4FBv**’s comments.
- We added a discussion on how our algorithm could be employed in real-life scenarios by leveraging offline learning and pre-training in our simulation environment, addressing concerns from reviewers **cToo** and **SXeV**. This discussion appears in Section 6 (Discussion and Conclusions).
- We have clarified our notion of merit-based equity in Section 3.2. This addresses the concern of reviewer **3gTN**.

We are pleased that the reviewers recognized the importance of our proposed solution and the effectiveness of our revisions to further highlight the suitability of our work, even in complex, real-world applications.

---

### Meta-Review · Area_Chair_Fi4N · 2026-01-09

**Summary:**

The reviewers acknowledge the importance of the problem tacked in the paper and the proposed method. However, during the review process, they also raised many and important concerns about the paper. Some of the main concern were related to i) the distinction with related works; ii) lack of theoretical results; iii) limited and unrealistic empirical evaluation; and iv) unclear assumptions. Most of these concerns have been partially addressed during the rebuttal, e.g., by extending description of related work and adding results on the German dataset. However, while I agree with reviews cToo on the potential significance of the paper,  I still believe that the changes needed in the paper are major and still in progress at the time of the end of the discussion period. I have looked myself to the revised version and I believe that the paper still need another round of reviews and improvements. Given the lack of theoretical results, I encourage the authors to significantly improve their empirical evaluation and show when and how the proposed method provides any advantages over existing approaches. Also, I do still have major concerns about the practical applicability of the proposed approach--- I remain unconvinced about the selection of the the time horizon T, as well as how realistic is to consider thousands of episodes before convergence.

In summary, the line of work has potential but I do believe the paper is not hyet ready for publication.

**Reviewer Concerns:**

I believe the paper has potential but it is not yet ready for publication. Specifically, the results on German now in the appendix seem very limited and do not show improvement with respect to "Hybrid" to really observed the advantages of the proposed method. See above for more details.

**Reviewer Scores:**

Some of the reviewers said that they increase their score, without making explicit claims about their support to acceptance.

---

### Decision · Program_Chairs · 2026-01-26

Reject